# Federated learning enables big data for rare cancer boundary detection

Although machine learning (ML) has shown promise across disciplines, out-of-sample generalizability is concerning. This is currently addressed by sharing multi-site data, but such centralization is challenging/infeasible to scale due to various limitations. Federated ML (FL) provides an alternative paradigm for accurate and generalizable ML, by only sharing numerical model updates. Here we present the largest FL study to-date, involving data from 71 sites across 6 continents, to generate an automatic tumor boundary detector for the rare disease of glioblastoma, reporting the largest such dataset in the literature ($n = 6, 314$). We demonstrate a 33% delineation improvement for the surgically targetable tumor, and 23% for the complete tumor extent, over a publicly trained model. We anticipate our study to: 1) enable more healthcare studies informed by large diverse data, ensuring meaningful results for rare diseases and underrepresented populations, 2) facilitate further analyses for glioblastoma by releasing our consensus model, and 3) demonstrate the FL effectiveness at such scale and task-complexity as a paradigm shift for multi-site collaborations, alleviating the need for data-sharing.

Recent technological advancements in healthcare, coupled with patients' culture shifting from reactive to proactive, have resulted in a radical growth of primary observations generated by health systems. This contributes to the burnout of clinical experts, as such observations require thorough assessment. To alleviate this situation, there have been numerous efforts for the development, evaluation, and eventual clinical translation of machine learning (ML) methods to identify relevant relationships among these observations, thereby reducing the burden on clinical experts. Advances in ML, and particularly deep learning (DL), have shown promise in addressing these complex healthcare problems. However, there are concerns about their generalizability on data from sources that did not participate in model training, i.e., "out-of-sample" data[1,2]. Literature indicates that training robust and accurate models requires large amounts of data[3–5], the diversity of which affects model generalizability to "out-of-sample" cases[6]. To address these concerns, models need to be trained on data originating from numerous sites representing diverse population samples. The current paradigm for such multi-site collaborations is "centralized learning" (CL), in which data from different sites are shared to a centralized location following inter-site agreements[6–9].

However, such data centralization is difficult to scale (and might not even be feasible), especially at a global scale, due to concerns[10,11] relating to privacy, data ownership, intellectual property, technical challenges (e.g., network and storage limitations), as well as compliance with varying regulatory policies (e.g., Health Insurance Portability and Accountability Act (HIPAA) of the United States[12] and the General Data Protection Regulation (GDPR) of the European Union[13]). In contrast to this centralized paradigm, "federated learning" (FL) describes a paradigm where models are trained by only sharing model parameter updates from decentralized data (i.e., each site retains its data locally)[10,11,14–16], without sacrificing performance when compared to CL-trained models[11,15,17–21]. Thus, FL can offer an alternative to CL, potentially creating a paradigm shift that alleviates the need for data sharing, and hence increase access to geographically distinct collaborators, thereby increasing the size and diversity of data used to train ML models.

FL has tremendous potential in healthcare[22,23], particularly towards addressing health disparities, under-served populations, and "rare" diseases[24], by enabling ML models to gain knowledge from ample and diverse data that would otherwise not be available. With

✉ e-mail: sbakas@upenn.edu

that in mind, here we focus on the "rare" disease of glioblastoma, and particularly on the detection of its extent using multi-parametric magnetic resonance imaging (mpMRI) scans[25]. While glioblastoma is the most common malignant primary brain tumor[26–28], it is still classified as a "rare" disease, as its incidence rate (i.e., 3/100,000 people) is substantially lower than the rare disease definition rate (i.e., <10/100,000 people)[24]. This means that single sites cannot collect large and diverse datasets to train robust and generalizable ML models, necessitating collaboration between geographically distinct sites. Despite extensive efforts to improve the prognosis of glioblastoma patients with intense multimodal therapy, their median overall survival is only 14.6 months after standard-of-care treatment, and 4 months without treatment[29]. Although the subtyping of glioblastoma has been improved[30] and the standard-of-care treatment options have expanded during the last 20 years, there have been no substantial improvements in overall survival[31]. This reflects the major obstacle in treating these tumors which is their intrinsic heterogeneity[26,28], and the need for analyses of larger and more diverse data toward a better understanding of the disease. In terms of radiologic appearance, glioblastomas comprise of three main sub-compartments, defined as (i) the "enhancing tumor" (ET), representing the vascular blood-brain barrier breakdown within the tumor, (ii) the "tumor core" (TC), which includes the ET and the necrotic (NCR) part, and represents the surgically relevant part of the tumor, and (iii) the "whole tumor" (WT), which is defined by the union of the TC and the peritumoral edematous/infiltrated tissue (ED) and represents the complete tumor extent relevant to radiotherapy (Fig. 1b). Detecting these sub-compartment boundaries, therefore, defines a multi-parametric multi-class learning problem and is a critical first step towards further quantifying and assessing this heterogeneous rare disease and ultimately influencing clinical decision-making.

Co-authors in this study have previously introduced FL in healthcare in a simulated setting[15] and further conducted a thorough quantitative performance evaluation of different FL workflows[11] (refer to supplementary figures for illustration) for the same use-case as the present study, i.e., detecting the boundaries of glioblastoma sub-compartments. Findings from these studies supported the superiority of the FL workflow used in the present study (i.e., based on an aggregation server[10,14]), which had almost identical performance to CL, for this use-case. Another study[32] has explored the first real-world federation for a breast cancer classification task using 5 sites, and another[16] used electronic medical records along with x-ray images from 20 sites to train a classifier to output a label corresponding to future oxygen requirement for COVID-19 patients.

This study describes the largest to-date global FL effort to develop an accurate and generalizable ML model for detecting glioblastoma sub-compartment boundaries, based on data from 6314 glioblastoma patients from 71 geographically distinct sites, across six continents (Fig. 1a). Notably, this describes the largest and most diverse dataset of glioblastoma patients ever considered in the literature. It was the use of FL that successfully enabled our ML model to gain knowledge from such an unprecedented dataset. The extended global footprint and the task complexity are what sets this study apart from current literature, since it dealt with a multi-parametric multi-class problem with reference standards that require expert clinicians following an involved manual annotation protocol, rather than simply recording a categorical entry from medical records[16,32]. Moreover, varying characteristics of the mpMRI data due to scanner hardware and acquisition protocol differences[33,34] were handled at each collaborating site via established harmonized preprocessing pipelines[35–39].

The scientific contributions of this manuscript can be summarized by (i) the insights garnered during this work that can pave the way for more successful FL studies of increased scale and task complexity, (ii) making a potential impact for the treatment of the rare disease of glioblastoma by publicly releasing clinically deployable trained consensus models, and most importantly, iii) demonstrating the effectiveness of FL at such scale and task complexity as a paradigm shift redefining multi-site collaborations, while alleviating the need for data sharing.

## Results

The complete federation followed a staged approach, starting from a "public initial model" (trained on data of 231 cases from 16 sites), followed by a "preliminary consensus model" (involving data of 2471 cases from 35 sites), to conclude on the "final consensus model" (developed on data of 6314 cases from 71 sites). To quantitatively evaluate the performance of the trained models, 20% of the total cases contributed by each participating site were excluded from the model training process and used as "local validation data". To further evaluate the generalizability of the models in unseen data, 6 sites were not involved in any of the training stages to represent an unseen "out-of-sample" data population of 590 cases. To facilitate further evaluation without burdening the collaborating sites, a subset ($n = 332$) of these cases was aggregated to serve as a "centralized out-of-sample" dataset. The training was initiated from a pre-trained model (i.e., our public initial model) rather than a random initialization point, in order to have faster convergence of the model performance[40,41]. Model performance was quantitatively evaluated here using the Dice similarity coefficient (DSC), which assesses the spatial agreement between the model's prediction and the reference standard for each of the three tumor sub-compartments (ET, TC, WT).

### Increased data can improve performance

When the federation began, the public initial model was evaluated against the local validation data of all sites, resulting in an average (across all cases of all sites) DSC per sub-compartment, of $DSC_{ET} = 0.63$, $DSC_{TC} = 0.62$, $DSC_{WT} = 0.75$. To summarize the model performance with a single collective score, we then calculate the average DSC (across all 3 tumor sub-compartments per case, and then across all cases of all sites) as equal to 0.66. Following model training across all sites, the final consensus model garnered significant performance improvements against the collaborators' local validation data of 27% ($p_{ET} < 1 \times 10^{-36}$), 33% ($p_{TC} < 1 \times 10^{-59}$), and 16% ($p_{WT} < 1 \times 10^{-21}$), for ET, TC, and WT, respectively (Fig. 1c). To further evaluate the potential generalizability improvements of the final consensus model on unseen data, we compared it with the public initial model against the complete out-of-sample data and noted significant performance improvements of 15% ($p_{ET} < 1 \times 10^{-5}$), 27% ($p_{TC} < 1 \times 10^{-16}$), and 16% ($p_{WT} < 1 \times 10^{-7}$), for ET, TC, and WT, respectively (Fig. 1d). Notably, the only difference between the public initial model and the final consensus model, was that the latter gained knowledge during training from increased datasets contributed by the complete set of collaborators. The conclusion of this finding reinforces the importance of using large and diverse data for generalizable models to ultimately drive patient care.

### Data size alone may not predict success

This is initially observed in our federated setting, where the comparative evaluation of the public initial model, the preliminary consensus model, and the final consensus model, against the centralized out-of-sample data, indicated performance improvements not directly related to the amount of data used for training. Specifically, we noted major significant ($p < 7 \times 10^{-18}$, Wilcoxon signed-rank test) performance improvements between the public initial model and the preliminary consensus model, as opposed to the insignificant ($p > 0.067$, Wilcoxon signed-rank test) ones between the preliminary and the final consensus model, as quantified in the centralized out-of-sample data for all sub-compartments and their average (Fig. 2).

We further expanded this analysis to assess this observation in a non-federated configuration, where we selected the largest collaborating sites (comprehensive cancer centers contributing > 200 cases,

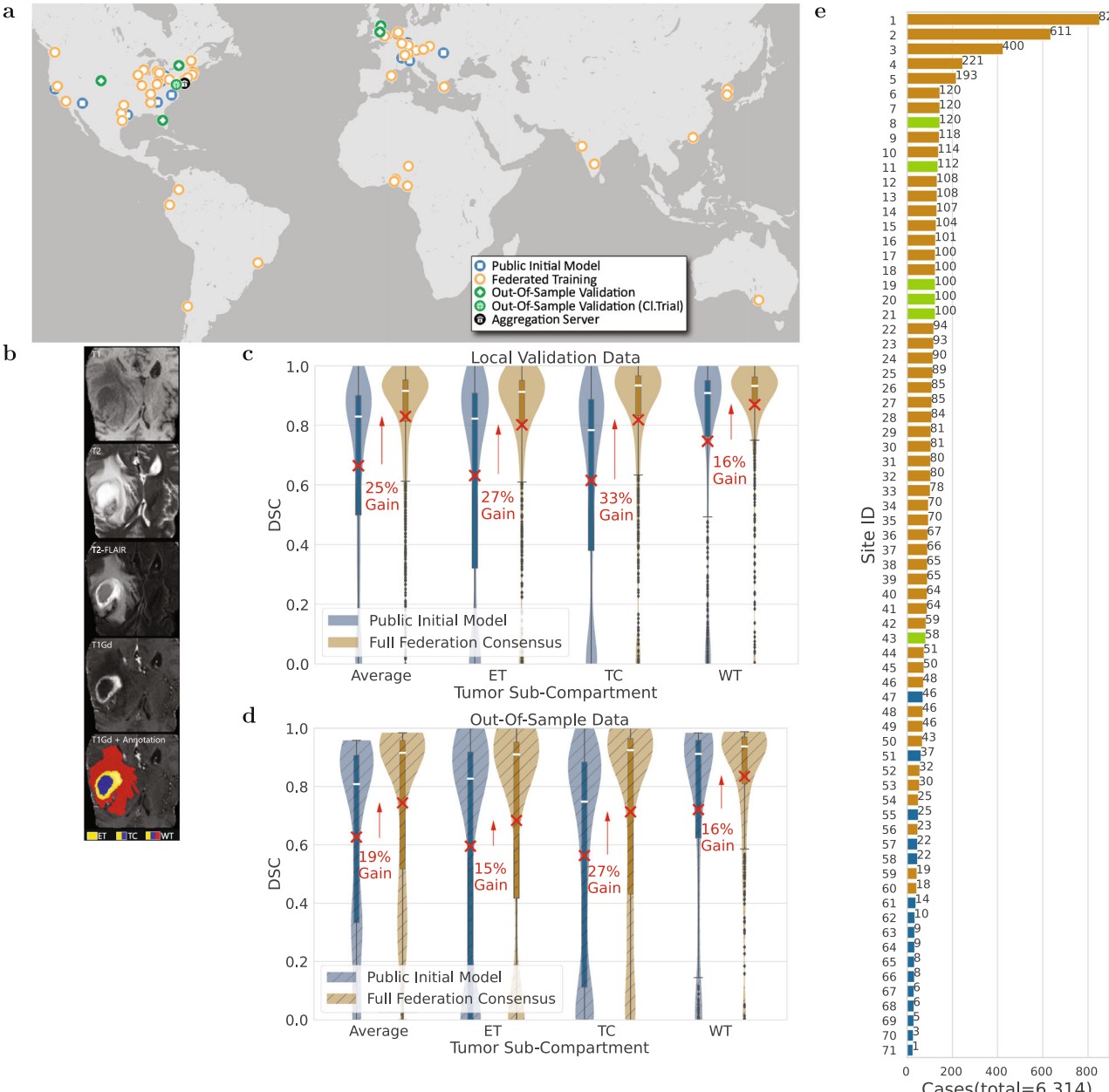

**Fig. 1 | Representation of the study's global scale, diversity, and complexity.**
**a** The map of all sites involved in the development of FL consensus model.
**b** Example of a glioblastoma mpMRI scan with corresponding reference annotations of the tumor sub-compartments (ET enhancing tumor, TC tumor core, WT whole tumor). **c, d** Comparative Dice similarity coefficient (DSC) performance evaluation of the final consensus model with the public initial model on the collaborators' local validation data (in **c** with $n = 1043$ biologically independent cases) and on the complete out-of-sample data (in **d** with $n = 518$ biologically independent cases), per tumor sub-compartment (ET enhancing tumor, TC tumor core, WT whole tumor). Note the box and whiskers inside each violin plot represent the true

min and max values. The top and bottom of each "box" depict the 3rd and 1st quartile of each measure. The white line and the red '×', within each box, indicate the median and mean values, respectively. The fact that these are not necessarily at the center of each box indicates the skewness of the distribution over different cases. The "whiskers" drawn above and below each box depict the extremal observations still within 1.5 times the interquartile range, above the 3rd or below the 1st quartile. Equivalent plots for the Jaccard similarity coefficient (JSC) can be observed in supplementary figures. **e** Number of contributed cases per collaborating site.

and familiar with computational analyses), and coordinated independent model training for each, starting from the public initial model and using only their local training data. The findings of this evaluation indicate that the final consensus model performance is always superior or insignificantly different ($p_{\text{Average}} = 0.1$, $p_{\text{ET}} = 0.5$, $p_{\text{TC}} = 0.2$, $p_{\text{WT}} = 0.06$, Wilcoxon signed-rank test) to the ensemble of the local models of these four largest contributing collaborators, for all tumor sub-compartments (Fig. 2). This finding highlights that even large sites can benefit from collaboration.

## FL is robust to data quality issues

Data quality issues relating to erroneous reference annotations (with potential negative downstream effects on output predictions) were identified by monitoring the global consensus model performance during training. However, only data quality issues that largely affected the global validation score could be identified and corrected during training. Those with more subtle effects in the global validation score were only identified after the completion of the model training by looking for relatively low local validation scores of the consensus

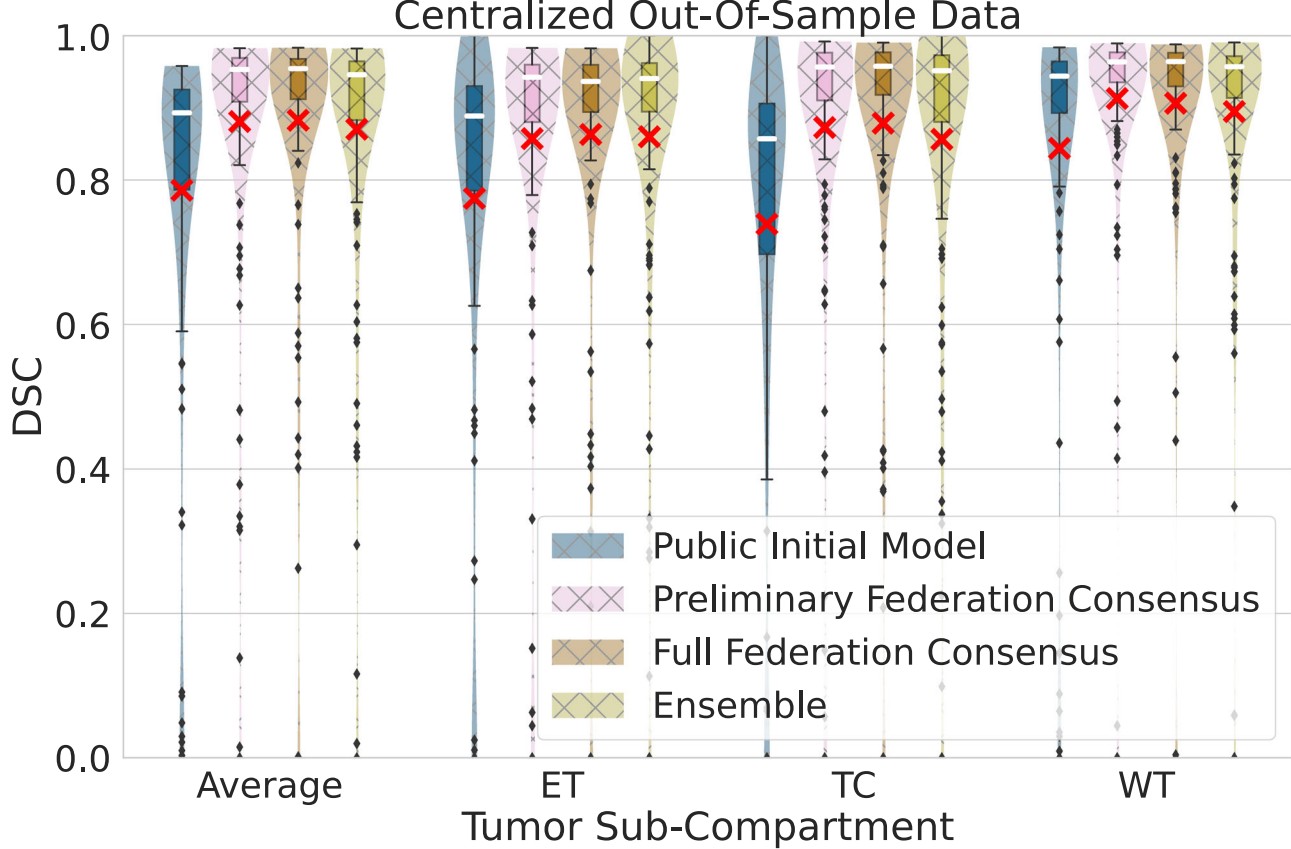

**Fig. 2 | Generalizable Dice similarity coefficient (DSC) evaluation on 'centralized' out-of-sample data (n = 154 biologically independent cases), per tumor sub-compartment (ET enhancing tumor, TC tumor core, WT whole tumor) and averaged across cases.** Comparative performance evaluation across the public initial model, the preliminary consensus model, the final consensus model, and an ensemble of single site models from collaborators holding > 200 cases. Note the box and whiskers inside each violin plot, represent the true min and max values. The top and bottom of each "box" depict the 3rd and 1st quartile of each measure. The white line and the red '×', within each box, indicate the median and mean values, respectively. The fact that these are not necessarily at the center of each box indicates the skewness of the distribution over different cases. The "whiskers" drawn above and below each box depict the extremal observations still within 1.5 times the interquartile range, above the 3rd or below the 1st quartile. Equivalent plots for Jaccard similarity coefficient (JSC) can be observed in supplementary figures.

model across collaborating sites. An example of such a quality issue with erroneous reference labels (from Site 48) is shown in Fig. 3c. Looking closer, local validation scores at Site 48 (Fig. 3b) are significantly different ($p_{ET} < 3 \times 10^{-12}$, $p_{TC} < 3 \times 10^{-12}$, $p_{WT} < 3 \times 10^{-12}$, Wilcoxon signed-rank test) than the average scores across the federation (Fig. 3a). Significant differences were calculated by sample pairs for each federated round, where a sample pair consists of the mean validation score over samples for Site 48 paired with those across all sites. These local validation scores (Fig. 3b) indicate that the model is not gaining knowledge from these local data, and their comparison with the average scores across the federation (Fig. 3a) indicates that the global consensus model performance is not adversely affected. This finding supports the importance of robustness at a global scale.

### FL benefits the more challenging tasks

The complexity of boundary detection drops when moving from smaller to larger sub-compartments, i.e., from ET to TC, and then to WT[35–38]. This is further confirmed here, as evidenced by the model's relative performance indicated by the local validation curves and their underlying associated areas in Fig. 3.a. Since the current clinically actionable sub-compartments are TC (i.e., considered for surgery) and WT (i.e., considered for radiotherapy)[42], performance improvements of their boundary detection may contribute to the model's clinical impact and relevance.

Our findings indicate that the benefits of FL are more pronounced for the more challenging sub-compartments, i.e., larger performance improvements for ET and TC compared to WT (Fig. 1c). Notably, the largest and most significant improvement (33%, $p < 7 \times 10^{-60}$) is noted for the TC sub-compartment, which is surgically actionable and not a trivial sub-compartment to delineate accurately[43,44]. This finding of FL benefiting the more challenging tasks rather than boosting performance on the relatively easier task (e.g., thresholding the abnormal T2-FLAIR signal for the WT sub-compartment) by gaining access to larger amounts of good quality data holds a lot of promise for FL in healthcare.

### Optimal model selection is non-trivial

Using the performance of the global consensus model during training across all local validation cases, two distinct model configurations were explored for selecting the final consensus model. Analyzing the sequence of consensus models produced during each federated round, we selected four different models: the *singlet*, for which the average DSC across all sub-compartments scored high, and three independent models, each of which yielded high *DSC* scores for each tumor sub-compartment, i.e., ET, TC, WT. We defined the collection of these three independent consensus models as a *triplet*.

To identify the best model, 5 *singlets* and 5 *triplets* were selected based on their relative performance on all local validation cases and

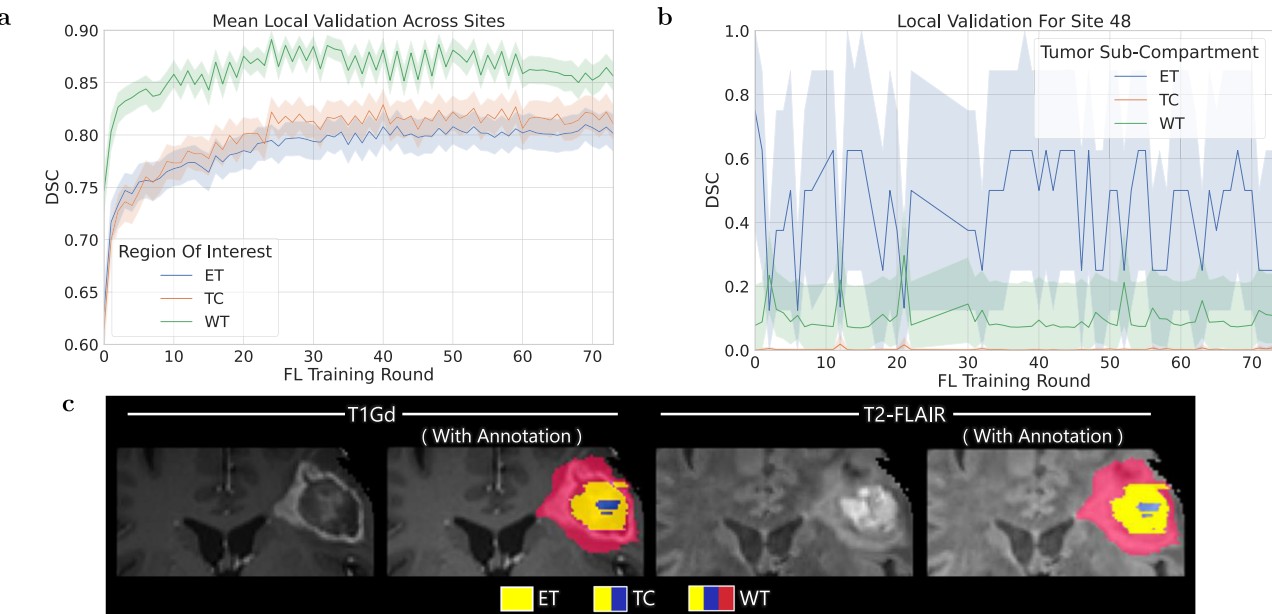

**Fig. 3 | Per-tumor region (ET enhancing tumor, TC tumor core, WT whole tumor) mean Dice similarity coefficient (*DSC*) over validation samples (with shading indicating 95% confidence intervals again over samples). a** At all participating sites across training rounds showing that the score is greater for sub-compartments with larger volumes. **b** For a site with problematic annotations (Site 48). The instability in these curves could be caused by errors in annotation for the local validation data (similar to errors that were observed for a small shared sample of data from this site). **c** Provides an example of a case with erroneous annotations in the data used by Site 48. Equivalent plots for Jaccard similarity coefficient (JSC) can be observed in supplementary figures.

evaluated against the centralized out-of-sample data. Only small differences are observed between the *singlet* and *triplet* models, and these differences diminish as the sub-compartment size increases. Comparing the means of *singlet* and *triplet*, the larger (and only significant) performance improvement difference compared to the public initial model is noted for the ET sub-compartment (improved by < 3%, $p_{ET} = 0.02$), followed by TC (improved by < 1.4%, $p_{TC} = 0.09$), and then lastly WT (improved by < 1.1%, $p_{WT} = 0.2$) (Tables S1 and S2). However, the decision of using a *singlet* or a *triplet* model should also rely on computational cost considerations, as *triplets* will be three times more expensive than *singlets* during model inference.

## Discussion

In this study, we have described the largest real-world FL effort to-date utilizing data of 6314 glioblastoma patients from 71 geographically unique sites spread across 6 continents, to develop an accurate and generalizable ML model for detecting glioblastoma sub-compartment boundaries. Notably, this extensive global footprint of the collaborating sites in this study also yields the largest dataset ever reported in the literature assessing this rare disease. It is the use of FL that successfully enabled (i) access to such an unprecedented dataset of the most common and fatal adult brain tumor, and (ii) meaningful ML training to ensure the generalizability of models across out-of-sample data. In comparison with the limited existing real-world FL studies[16,32], our use-case is larger in scale and substantially more complex, since it (1) addresses a multi-parametric multi-class problem, with reference standards that require expert collaborating clinicians to follow an involved manual annotation protocol, rather than simply recording a categorical entry from medical records, and (2) requires the data to be preprocessed in a harmonized manner to account for differences in MRI acquisition. Since glioblastoma boundary detection is critical for treatment planning and the requisite first step for further quantitative analyses, the models generated during this study have the potential to make a far-reaching clinical impact.

The large and diverse data that FL enabled, led to the final consensus model garnering significant performance improvements over the public initial model against both the collaborators' local validation data and the complete out-of-sample data. The improved result is a clear indication of the benefit that can be afforded through access to more data. However, increasing the data size for model training without considerations relating to data quality, reference labels, and potential site bias (e.g., scanner acquisition protocols, demographics, or sociocultural considerations, such as more advanced presentation of disease at diagnosis in low-income regions[45]) might not always improve results. Literature also indicates an ML performance stagnation effect, where each added case contributes less to the model performance as the number of cases increase[46]. This is in line with our finding in the federated setting (Fig. 2), where performance improvements across the public initial model, the preliminary consensus model, and the final consensus model, were not directly/linearly related to the amount of data used for training. This happened even though the final consensus model was trained on over twice the number of cases (and included 2 of the largest contributing sites—Sites 1 and 4) when compared to the preliminary consensus model. Further noting that the preliminary federation model was already within the intra- and inter-rater variability range for this use-case (20% and 28%, respectively)[47], any further improvements for the full federation consensus model would be expected to be minimal[35–38].

To further assess these considerations, we coordinated independent model training for the four largest collaborating sites (i.e., >200 cases) by starting from the same public initial model and using only their local training data. The ensemble of these four largest site local models did not show significant performance differences to the final consensus model for any tumor sub-compartment, yet the final consensus model showed superior performance indicating that even sites with large datasets can benefit from collaboration. The underlying assumption for these results is that since each of these collaborators initiated their training from the public initial model (which included diverse data from 16 sites), their independent models and their ensemble could have inherited some of the initial model's data diversity, which could justify the observed insignificant differences (Fig. 2 and Supplementary Fig. 3). Though these findings are an indication

that the inclusion of more data alone may not lead to better performance, it is worth noting that these four largest sites used for the independent model training represent comprehensive cancer centers (compared to hospitals in community settings) with affiliated sophisticated labs focusing on brain tumor research, and hence were familiar with the intricacies of computational analyses. Further considering the aforementioned ML performance stagnation effect, we note the need for generalizable solutions to quantify the contribution of collaborating sites to the final consensus model performance, such that future FL studies are able to formally assess both the quantity and the quality of the contributed data needed by the collaborating sites and decide on their potential inclusion on use-inspired studies.

As noted in our results, due to the lack of such generalizable solutions, we were only able to identify quality issues after the model training. Specifically, we hypothesize that although Site 48 had data quality issues, its effect on the consensus model performance was not significant due to its relatively small dataset ($n = 46$) when compared to the other collaborating sites. The curves of Fig. 3a indicate that the global consensus model continues to consistently gain knowledge from the federation as a whole during training, highlighting robustness to such data quality issues. It remains unknown, however, how much better the consensus model would have performed if sites with problematic data were excluded or if these specific problematic data at Site 48 were excluded or corrected. These findings are aligned with literature observations (on the same use-case)[48], where a DL model[49] trained on 641 glioblastoma cases from 8 sites produced higher quality predictions on average than those created as reference standard labels by radiology expert operators. Quality was judged by 20 board-certified neuroradiologists, in a blinded side-by-side comparison of 100 sequestered unseen cases, and concluded that perfect or near-perfect reference labels may not be required to produce high-quality prediction systems. In other words, DL models may learn to see past imperfect reference training labels. These findings provide the impetus for further experimentation as they have implications for future FL studies. Future research is needed to automatically detect anomalies in the consensus model performance during training, particularly associated with contributions from individual sites.

There are a number of practical considerations that need to be taken into account to set up a multi-national real-world federation, starting with a substantial amount of coordination between each participating site. As this study is the first at this scale and task complexity, we have compiled a set of governance insights from our experience that can serve as considerations for future successful FL studies. These insights differ from previous literature that describes studies that were smaller in scale and involved simpler tasks[16,32]. By "governance" of the federation we refer both to the accurate definition of the problem statement (including reference labels and harmonization considerations accounting for inter-site variability), and the coordination with the collaborating sites for eligibility and compliance with the problem statement definition, as well as security and technical considerations. For future efforts aiming to conduct studies of a similar global scale, it would be beneficial to identify a solution for governance prior to initiating the study itself.

The coordination began with engaging the security teams of collaborating sites and providing them access to the source code of the platform developed to facilitate this study. These security discussions highlighted the benefit of the platform being open-source, making security code reviews easier. Resource gathering was then carried out by identifying technical leads and assessing computational resources at each site. With the technical leads, we then proceeded to test the complete workflow to further identify gaps in the requirements, such as network configurations and hardware requirements. We then proceeded with data curation and preprocessing, and finally connected individual sites to the aggregation server to initiate their participation.

Following the precise definition of our problem statement[35–38], ensuring strict compliance with the preprocessing and annotation protocol for the generation of reference standards was vital for the model to learn correct information during training. To this end, we instituted an extensively and comprehensively documented annotation protocol with visual example representations and common expected errors (as observed in the literature[38,50]) to all collaborators. We have further circulated an end-to-end platform[39] developed to facilitate this federation, providing to each collaborating site all the necessary functionalities to (i) uniformly curate their data and account for inter-site acquisition variability, (ii) generate the reference standard labels, and (iii) participate in the federated training process. Finally, we held interactive sessions to complement the theoretical definition of the reference standards, and further guide collaborating sites. Particular pain points regarding these administrative tasks included managing the large volume of communication (i.e., emails and conference calls) needed to address questions and issues that arose, as well as the downtime incurred in FL training due to issues that had not yet been identified and were adversely affecting the global model. Though we developed many ad-hoc tools for this workflow ourselves (particularly for the data processing and orchestration steps), many issues we encountered were common enough in retrospect (for example common Transport Layer Security (TLS) errors) that mature automated solutions will address them. Many of these automations will be use-case dependent, such as the MRI data corruption checks we used from the FeTS tool[39]. For these use-case-dependent automation, more associated tools are expected to become available as various domain experts enter into the FL community, while some will be more general purpose. As our inspection of both local and global model validation scores was manual during our deployment, we in retrospect see great value in automated notifications (performed at the collaborator infrastructure to help minimize data information leakage) to alert a collaborator (or the governor) when their local or global model validation is significantly low. Such an alert can indicate the potential need to visually inspect example failure cases in their data for potential issues. With continued efforts towards developing automated administration tools around FL deployments, we expect the coordination for large FL deployments to become easier.

In general, debugging issues with the inputted local data and annotations is more difficult during FL due to the level of coordination and/or privacy issues involved, since the data are always retained at the collaborating site. We gained substantial experience during this effort that went into further development of use-inspired but generalizable data sanity-checking functionality in the tools we developed, towards facilitating further multi-site collaborations.

Upon conclusion of the study, sites participating in the model training process were given a survey to fill in regarding various aspects of their experience. According to the provided feedback, 96% of the sites found the comprehensive documentation on preprocessing and data curation essential and thought that lack of such documentation could have resulted in inconsistent annotations. Additionally, 92% found the documentation relating to establishing secure connectivity to the aggregation server easy to follow and essential to expedite reviews by the related groups. Furthermore, 84% of the sites appreciated the user-friendly interface of the provided tool and its associated complete functionality (beyond its FL backend), and indicated their intention to use it and recommend it for projects and data analysis pipelines beyond the scope of this study. To generate the reference standard labels for their local data, 86% of the collaborating sites indicated that they used either the FeTS Tool[39] (i.e., the tool developed for this study), CaPTk[51], or ITK-SNAP[52], whereas the remaining 14% used either 3D-Slicer[53], the BraTS toolkit[54], or something else. In terms of hardware requirements at each site, 88% used a dedicated workstation for their local workload, and the remaining 12% used either a containerized form of the FeTS tool or a virtual machine.

Although data are always retained within the acquiring site during FL (and hence FL is defined as private-by-design), different security and privacy threats remain[55–57]. These threats include attempted extraction of training data information from intermediate and final models, model theft, and submission of poison model updates with the goal of introducing unwanted model behavior (including incentivizing the model to memorize more information about the training data in support of subsequent extraction, i.e., leakage). A number of technologies can be used to mitigate security and privacy concerns during FL[55–57]. Homomorphic encryption[58], secure multiparty compute[59], and trusted execution environments (TEEs)[60,61] allow for collaborative computations to be performed with untrusted parties while maintaining confidentiality of the inputs to the computation. Differentially private training algorithms[62–64] allow for mitigation of information leakage from both the collaborator model updates and the global consensus aggregated models. Finally, assurance that remote computations are executed with integrity can be designed for with the use of hardware-based trust provided by TEEs, as well as with some software-based integrity checking[65]. Each of these technologies comes with its own benefits in terms of security and/or privacy, as well as costs and limitations, such as increased computational complexity, associated hardware requirements and/or reduced quality of computational output (such as the reduction of model utility that can be associated with differentially private model training). Further experimentation needs to be done in order to best inform prospective federations as to which technologies to use towards addressing their specific concerns within the context of the collaborator infrastructure and trust levels, depending on the use-case, the extent of the collaborating network, and the level of trust within the involved parties. Our study was based on a collaborative network of trusted sites, where authentication was based on personal communication across collaborating sites and the combination of TLS and TEEs were considered sufficient.

Although our study has the potential to become the baseline upon which future ML research studies will be done, there is no automated mechanism to assess inputted data quality from collaborators, which could result in models trained using sub-optimal data. Additionally, we used a single off-the-shelf neural network architecture for training, but it has been shown that model ensembles perform better for the task at hand[35–38], and it remains to be explored how such a strategy could be explored in a federated study. Moreover, the instantiation of the federation involved a significant amount of coordination between each site and considering the limited real-world FL studies at the time, there were no tools available to automate such coordination and orchestration. These involved (i) getting interviewed by information security officers of collaborating sites, (ii) ensuring that the harmonized pre-processing pipeline was used effectively, (iii) clear communication of the annotation protocol, and iv) testing the network communication between the aggregator and each site. This amount of effort, if not aided by automated tools, will continue to be a huge roadblock for FL studies, and dedicated coordination and orchestration resources are required to conduct this in a reproducible and scalable manner.

We have demonstrated the utility of an FL workflow to develop an accurate and generalizable ML model for detecting glioblastoma sub-compartment boundaries, a finding which is of particular relevance for neurosurgical and radiotherapy planning in patients with this disease. This study is meant to be used as an example for future FL studies between collaborators with an inherent amount of trust that can result in clinically deployable ML models. Further research is required to assess privacy concerns in a detailed manner[63,64] and to apply FL to different tasks and data types[66–69]. Building on this study, a continuous FL consortium would enable downstream quantitative analyses with implications for both routine practice and clinical trials, and most importantly, increase access to high-quality precision care worldwide. Furthermore, the lessons learned from this study with such a global footprint are invaluable and can be applied to a broad array of clinical scenarios with the potential for great impact on rare diseases and underrepresented populations.

## Methods

The study and results presented in this manuscript comply with all relevant ethical regulations and follow appropriate ethical standards in conducting research and writing the manuscript, following all applicable laws and regulations regarding the treatment of human subjects. Use of the private retrospective data collection of each collaborating site has been approved by their respective institutional review board, where informed consent from all participants was also obtained and stored.

### Data

The data considered in this study described patient populations with adult-type diffuse glioma[30], and specifically displaying the radiological features of glioblastoma, scanned with mpMRI to characterize the anatomical tissue structure[25]. Each case is specifically described by (i) native T1-weighted (T1), (ii) Gadolinium-enhanced T1-weighted (T1Gd), (iii) T2-weighted (T2), and (iv) T2-weighted-Fluid-Attenuated-Inversion-Recovery (T2-FLAIR) MRI scans. Cases with any of these sequences missing were not included in the study. Note that no inclusion/exclusion criterion applied relating to the type of acquisition (i.e., both 2D axial and 3D acquisitions were included, with a preference for 3D if available), or the exact type of sequence (e.g., MP-RAGE vs. SPGR). The only exclusion criterion was for T1-FLAIR scans that were intentionally excluded to avoid mixing varying tissue appearance due to the type of sequence, across native T1-weighted scans.

The publicly available data from the International Brain Tumor Segmentation (BraTS) 2020 challenge[35–37], was used to train the public initial model of this study. The BraTS challenge[35–38], seeking methodological advancements in the domain of neuro-oncology, has been providing the community with (i) the largest publicly available and manually-curated mpMRI dataset of diffuse glioma patients (an example of which is illustrated in Fig. 1b), and (ii) a harmonized pre-processing pipeline[51,70,71] to handle differences in inter-site acquisition protocols. The public initial model was used to initialize the FL training, instead of a randomly generated initialization, as starting from a pre-trained model leads to faster convergence[41]. The complete BraTS 2020 dataset originally included cases from sites that also participated in this study as independent collaborators. To avoid any potential data leakage, we reduced the size of the complete BraTS dataset by removing cases acquired by these specific sites, resulting in a dataset of 231 cases from 16 international sites, with varying contributing cases across sites (Fig. 1e). The exact site IDs that construct the data of the public initial model are: 47, 51, 55, 57, 58, 61, 62, 63, 64, 65, 66, 67, 68, 69, 70, and 71. Subsequently, the resulting dataset was split at a 4:1 ratio between cases for training ($n = 185$) and validation ($n = 46$).

The eligibility of collaborating sites to participate in the federation was determined based on data availability, and approval by their respective institutional review board. 55 sites participated as independent collaborators in the study defining a dataset of 6083 cases. The MRI scanners used for data acquisition were from multiple vendors (i.e., Siemens, GE, Philips, Hitachi, Toshiba), with magnetic field strength ranging from 1T to 3T. The data from all 55 collaborating sites followed a male:female ratio of 1.47:1 with ages ranging between 7 and 94 years.

From all 55 collaborating sites, 49 were chosen to be part of the training phase, and 6 sites were categorized as "out-of-sample", i.e., none of these were part of the training stage. These specific 6 out-of-sample sites (Site IDs: 8, 11, 19, 20, 21, 43) were allocated based on their availability, i.e., they have indicated expected delayed participation rendering them optimal for model generalizability validation. One of these 6 out-of-sample sites (Site 11) contributed aggregated a priori data from a multi-site randomized clinical trial for newly diagnosed

glioblastoma (ClinicalTrials.gov Identifier: NCT00884741, RTOG0825[72,73], ACRIN6686[74,75]), with inherent diversity benefiting the intended generalizability validation purpose. The American College of Radiology (ACR - Site 11) serves as the custodian of this trial's imaging data on behalf of ECOG-ACRIN, which made the data available for this study. Following screening for the availability of the four required mpMRI scans with sufficient signal-to-noise ratio judged by visual observation, a subset of 362 cases from the original trial data were included in this study. The out-of-sample data totaled 590 cases intentionally held out of the federation, with the intention of validating the consensus model in completely unseen cases. To facilitate further such generalizability evaluation without burdening the collaborating sites, a subset consisting of 332 cases (including the multi-site clinical data provided by ACR) from this out-of-sample data was aggregated, to serve as the "centralized out-of-sample" dataset. Furthermore, the 49 sites participating in the training phase define a collective dataset of 5493 cases. The exact 49 site IDs are: 1, 2, 3, 4, 5, 6, 7, 9, 10, 12, 13, 14, 15, 16, 17, 18, 22, 23, 24, 25, 26, 27, 28, 29, 30, 31, 32, 33, 34, 35, 36, 37, 38, 39, 40, 41, 42, 44, 45, 46, 48, 49, 50, 52, 53, 54, 56, 59, 60. These cases were automatically split at each site following a 4:1 ratio between cases for training and local validation. During the federated training phase, the data used for the public initial model were also included as a dataset from a separate node, such that the contribution of sites providing the publicly available data is not forgotten within the global consensus model. This results in the final consensus model being developed based on data from 71 sites over a total dataset of 6314 cases. Collective demographic information of the included population is provided in Table S3.

## Harmonized data preprocessing

Once each collaborating site identified its local data, they were asked to use the preprocessing functionality of the software platform we provided. This functionality follows the harmonized data preprocessing protocol defined by the BraTS challenge[35–38], as described below. This would allow accounting for inter-site acquisition protocol variations, e.g., 3D vs. 2D axial plane acquisitions.

**File-type conversion/patient de-identification.** The respective mpMRI scans (i.e., T1, T1Gd, T2, T2-FLAIR) of every case are downloaded onto a local machine in the Digital Imaging and Communications in Medicine (DICOM) format[76–78] and converted to the Neuroimaging Informatics Technology Initiative (NIfTI) file format[79] to ensure easier parsing of the volumetric scans during the computational process. The conversion of DICOM to NIfTI files has the benefit of eliminating all patient-identifiable metadata from the header portion of the DICOM format[80,81].

**Rigid registration.** Once the scans are converted to the NIfTI format, each volume is registered to a common anatomical space, namely the SRI24 atlas[82], to ensure a cohesive data shape ([240, 240, 155]) and an isotropic voxel resolution (1 mm³), thereby facilitating in the tandem analysis of the mpMRI scans. One of the most common types of MRI noise is based on the inhomogeneity of the magnetic field[83]. It has been previously[36] shown that the use of non-parametric, non-uniform intensity normalization to correct for these bias fields[84,85] obliterates the MRI signal relating to the regions of abnormal T2-FLAIR signal. Here, we have taken advantage of this adverse effect and used the bias field-corrected scans to generate a more optimal rigid registration solution across the mpMRI sequences. The bias field-corrected images are registered to the T1Gd image, and the T1Gd image is rigidly registered to the SRI24 atlas, resulting in two sets of transformation matrices per MRI sequence. These matrices are then aggregated into a single matrix defining the transformation of each MRI sequence from its original space to the atlas. We then apply this single aggregated matrix to

the NIfTI scans prior to the application of the bias field correction to maximize the fidelity of the finally registered images.

**Brain extraction.** This process focuses on generating a brain mask to remove all non-brain tissue from the image (including neck, fat, eyeballs, and skull), to enable further computational analyses while avoiding any potential face reconstruction/recognition[86]. For this step we utilized the Brain Mask Generator (BrainMaGe)[87], which has been explicitly developed to address brain scans in presence of diffuse glioma and considers brain shape as a prior, hence being agnostic to the sequence/modality input.

## Generation of automated baseline delineations of tumor sub-compartment boundaries.

We provided the ability to the collaborating sites to generate automated delineations of the tumor sub-compartments from three popular methods from the BraTS challenge, using models trained using the challenge's training data: (i) DeepMedic[49], (ii) DeepScan[88], and (iii) nnU-Net[89]. Along with segmentations from each method, label fusion strategies were also employed to provide a reasonable approximation to the reference labels that should be manually refined and approved by expert neuroradiologists to create the final reference labels. The label fusion approaches considered were i) standard voting[90], (ii) Simultaneous Truth And Performance Level Estimation (STAPLE)[91,92], iii) majority voting[93], and iv) Selective and Iterative Method for Performance Level Estimation (SIMPLE)[94].

**Manual refinements towards reference standard labels.** It was communicated to all participating sites to leverage the annotations generated using the automated mechanism as a baseline on which manual refinements were needed by neuroradiology experts, following a consistently communicated annotation protocol. The reference annotations comprised the Gd-enhancing tumor (ET−label '4'), the peritumoral edematous/invaded tissue (ED−label '2'), and the necrotic tumor core (NCR−label '1'). ET is generally considered the most active portion of the tumor, described by areas with both visually avid, as well as faintly avid, enhancement on the T1Gd scan. NCR is the necrotic part of the tumor, the appearance of which is hypointense on the T1Gd scan. ED is the peritumoral edematous and infiltrated tissue, defined by the abnormal hyperintense signal envelope on the T2-FLAIR scans, which includes the infiltrative non-enhancing tumor, as well as vasogenic edema in the peritumoral region[35–38] (an illustration can be seen in Fig. 1b).

**Data splits.** Once the data were preprocessed, training and validation cohorts were created randomly in a 4:1 ratio, and the splits were preserved during the entire duration of the FL training to prevent data leakage. The performance of every model was compared against the local validation data cohort on every federated round.

## Data loading and processing

We leveraged the data loading and processing pipeline from the Generally Nuanced Deep Learning Framework (GaNDLF)[95], to enable experimentation with various data augmentation techniques. Immediately after data loading, we removed the all-zero axial, coronal, and sagittal planes from the image, and performed a z-score normalization of the non-zero image intensities[96]. Each tumor sub-compartment of the reference label is first split into an individual channel and then passed to the neural network for processing. We extracted a single random patch per mpMRI volume set during every federated round. The patch size was kept constant at [128, 128, 128] to ensure that the trained model can fit the memory of the baseline hardware requirement of each collaborator, i.e., a discrete graphics processing unit with a minimum of 11 GB dedicated memory. For data augmentation, we added random noise augmentation ($\mu = 0.0$, $\sigma = 0.1$) with a probability

of $p = 0.2$, random rotations (90° and 180°, with the axis of rotation being uniformly selected in each case from the set of coronal, sagittal, and axial planes) each with a probability of $p = 0.5$, and a random flip augmentation with a probability of $p = 1.0$ with equal likelihood of flips across the sagittal, coronal, and axial planes.

## The neural network architecture

The trained model to delineate the different tumor sub-compartments was based on the popular 3D U-Net with residual connections (3D-ResUNet)[97–101], an illustration of which can be seen in the Supplementary Fig. 1. The network had 30 base filters, with a learning rate of lr = $5 \times 10^{-5}$ optimized using the Adam optimizer[102]. For the loss function used in training, we used the generalized DSC score[103,104] (represented mathematically in Eq. (1)) on the absolute complement of each tumor sub-compartment independently. Such mirrored DSC loss has been shown to capture variations in smaller regions better[89]. No penalties were used in the loss function, due to our use of 'mirrored' DSC loss[105–107]. The final layer of the model was a sigmoid layer, providing three channel outputs for each voxel in the input volume, one output channel per tumor sub-compartment. While the generalized *DSC* score was calculated using a binarized version of the output (check sigmoid value against the threshold 0.5) for the final prediction, we used the floating point *DSC*[108] during the training process.

$$DSC = \frac{2|RL \odot PM|_1}{|RL|_1 + |PM|_1} \qquad (1)$$

where RL serves as the reference label, PM is the predicted mask, $\odot$ is the Hadamard product[109] (i.e., component-wise multiplication), and $|x|_1$ is the L1-norm[110], i.e., the sum of the absolute values of all components).

## The Federation

The collaborative network of the present study spans 6 continents (Fig. 1), with data from 71 geographically distinct sites. The training process was initiated when each collaborator securely connected to a central aggregation server, which resided behind a firewall at the University of Pennsylvania. We have identified this FL workflow (based on a central aggregation server) as the optimal for this use-case, following a performance evaluation[11] for this very same task, i.e., detecting glioblastoma sub-compartment boundaries. As soon as the secure connection was established, the public initial model was passed to the collaborating site. Using FL based on an aggregation server (refer to supplementary figures for illustration), collaborating sites then trained the same network architecture on their local data for one epoch, and shared model updates with the central aggregation server. The central aggregation server received model updates from all collaborators, combined them (by averaging model parameters) and sent the consensus model back to each collaborator to continue their local training. Each such iteration is called a "federated round". Based on our previously conducted performance evaluation for this use-case[11], we chose to perform aggregation of all collaborator updates in the present study, using the federated averaging (FedAvg) approach[14], i.e., average of collaborator's model updates weighted according to collaborator's contributing data. We expect these aggregation strategy choices to be use-case dependent, by providing due consideration to the collaborators' associated compute and network infrastructure. In this study, all the network communications during the FL model training process were based on TLS[111], to mitigate potential exposure of information during transit. Additionally, we demonstrated the feasibility of TEEs[60,61] for federated training by running the aggregator workload on the secure enclaves of Intel's Secure Guard Extensions (SGX) hardware (Intel® Xeon® E-2286M vPro 8-Core 2.4-5.0GHz Turbo), which ensured the confidentiality of the updates being aggregated and the integrity of the consensus model. TLS and TEEs can help mitigate some of the security and privacy concerns that remain for FL[55]. After not observing any meaningful changes since round 42, we stopped the training after a total of 73 federated rounds. Additionally, we performed all operations on the aggregator on secure hardware (TEE[112]), in order to increase the trust by all parties in the confidentiality of the model updates being computed and shared, as well as to increase the confidence in the integrity of the computations being performed[113].

We followed a staged approach for the training of the global consensus model, starting from a preliminary smaller federation across a subset ($n = 35$) of the participating sites to evaluate the complete process and resolve any initial network issues. Note that 16 of these 35 sites were used to train the public initial model, and used in the preliminary federation as an aggregated dataset. The exact 19 site IDs that participated in the training phase of the preliminary federation, as independent sites are: 2, 3, 9, 14, 22, 23, 24, 27, 28, 29, 31, 33, 36, 37, 41, 46, 53, 54, and 59. The total data held by this smaller federation represented approximately 42% ($n = 2471$) of the data used in the full federation. We also trained individual models (initialized using the public initial model) using centralized training at all sites holding >200 training cases, and performed a comparative evaluation of the consensus model with an ensemble of these "single site models". The per voxel sigmoid outputs of the ensemble were computed as the average of such outputs over the individual single-site models. As with all other models in this study, binary predictions were computed by comparing these sigmoid outputs to a threshold value of 0.5. The single-site model ensemble utilized (via the data at the single site) approximately 33% of the total data across the federation.

## Model runtime in low-resource settings

Clinical environments typically have constrained computational resources, such as the availability of specialized hardware (e.g., DL acceleration cards) and increased memory, which affect the runtime performance of DL inference workloads. Thus, taking into consideration the potential deployment of the final consensus model in such low-resource settings, we decided to proceed with a single 3D-ResU-Net, rather than an ensemble of multiple models. This decision ensured a reduced computational burden when compared with running multiple models, which is typically done in academic research projects[35–38].

To further facilitate use in low-resource environments, we have provided a post-training run-time optimized[114] version of the final consensus model. Graph level optimizations (i.e., operators fusion) were initially applied, followed by optimizations for low precision inference, i.e., converting the floating point single precision model to a fixed precision 8-bit integer model (a process known as "quantization"[115]). In particular, we used accuracy-aware quantization[116], where model layers were iteratively scaled to a lower precision format. These optimizations yielded run-time performance benefits, such as lower inference latency (a platform-dependent 4.48 × average speedup and 2.29 × reduced memory requirement when compared with the original consensus model) and higher throughput (equal to the 4.48 × speedup improvement since the batch size used is equal to 1), while the trade-off was an insignificant ($p_{Average} < 7 \times 10^{-5}$) drop in the average *DSC*.

**Clinically-deployable consensus models.** To further encourage the reproducibility of our study, and considering enhancing the potential impact for the study of the rare disease of glioblastoma, we publicly released the trained models of this study. We specifically released the final *singlet* and *triplet* consensus models, including the complete source code used in the project. Taking into consideration the potential deployment of these models in clinical settings, we refrained from training an ensemble of models (as typically done in academic

research projects[35–38]), due to the additional computational burden of running multiple models. Furthermore, to facilitate use in low-resource environments, we also provide a post-training run-time optimized[114] version of the final consensus model that obviates the need for any specialized hardware (such as DL acceleration cards) and performs insignificantly different from the final consensus model when evaluated against the centralized out-of-sample data.

### Reporting summary
Further information on research design is available in the Nature Research Reporting Summary linked to this article.

## Data availability
The datasets used in this study, from the 71 participating sites, are not made publicly available as a collective data collection due to restrictions imposed by acquiring sites. The public initial model data from 16 sites are publicly available through the BraTS challenge[35–38] and are available from https://www.med.upenn.edu/cbica/brats2020. The data from each of the 55 collaborating sites were neither publicly available during the execution of the study, nor shared among collaborating sites or with the aggregator. They were instead used locally, within each of the acquiring sites, for the training and validation of the global consensus model at each federated round. The anatomical template used for co-registration during preprocessing is the SRI24 atlas[82] and is available from https://www.nitrc.org/projects/sri24.

Source data are provided with this paper. Specifically, we provide the raw data, the associated python scripts, and specific instructions to reproduce the plots of this study in a GitHub repository, at: github.com/FETS-AI/2022_Manuscript_Supplement. The file 'SourceData.tgz', in the top directory holds an archive of csv files representing the source data. The python scripts are provided in the 'scripts' folder which utilize these source data and save '.png' images to disc and/or print latex code (for tables) to stdout. Furthermore, we have provided three sample validation cases, from the publicly available BraTS dataset, to qualitatively showcase the segmentation differences (small, moderate, and large) across the final global consensus model, the public initial model, and the ground truth annotations in the same GitHub repository.

## Code availability
Motivated by findability, accessibility, interoperability, and reusability (FAIR) criteria in scientific research[117], all the code used to design the Federated Tumor Segmentation (FeTS) platform[118] for this study is available through the FeTS Tool[39] and it is available at github.com/FETS-AI/Front-End. The functionality related to preprocessing (i.e., DICOM to NIfTI conversion, population-based harmonized preprocessing, co-registration) and manual refinements of annotation is derived from the open-source Cancer Imaging Phenomics Toolkit (CaPTk, github.com/CBICA/CaPTk)[51,70,71]. The co-registration is performed using the Greedy framework[119], available via CaPTk[51,70,71], ITK-SNAP[52], and the FeTS Tool[39]. The brain extraction is done using the BrainMaGe method[87], and is available at github.com/CBICA/BrainMaGe, and via GaNDLF[95] at github.com/mlcommons/GaNDLF. To generate automated annotations, DeepMedic's[49] integration with CaPTk was used, and we used the model weights and inference mechanism provided by the other algorithm developers (DeepScan[88] and nnU-Net[89] (github.com/MIC-DKFZ/nnunet)). DeepMedic's original implementation is available in github.com/deepmedic/deepmedic, whereas the one we used in this study can be found at github.com/CBICA/deepmedic. The fusion of the labels was done using the Label Fusion tool[120] available at github.com/FETS-AI/LabelFusion. The data loading pipeline and network architecture were developed using the GaNDLF framework[95] by using PyTorch[121]. The data augmentation was done via GaNDLF by leveraging TorchIO[122]. The FL backend developed for this project has been open-sourced as a separate software library,

to encourage further research on FL[123] and is available at github.com/intel/openfl. The optimization of the consensus model inference workload was performed via OpenVINO[124] (github.com/openvinotoolkit/openvino/tree/2021.4.1), which is an open-source toolkit enabling acceleration of neural network models through various optimization techniques. The optimizations were evaluated on an Intel Core® i7-1185G7E CPU @ 2.80 GHz with 2 × 8 GB DDR4 3200 MHz memory on Ubuntu 18.04.6 OS and Linux kernel version 5.9.0-050900-generic.

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

## Acknowledgements

Research and main methodological developments reported in this publication were partly supported by the National Institutes of Health (NIH) under award numbers NIH/NCI:U01CA242871 (S. Bakas), NIH/NINDS:R01NS042645 (C. Davatzikos), NIH/NCI:U24CA189523 (C. Davatzikos), NIH/NCI:U24CA215109 (J. Saltz), NIH/NCI:U01CA248226 (P. Tiwari), NIH/NCI:P30CA51008 (Y. Gusev), NIH:R50CA211270 (M. Muzi), NIH/NCATS:UL1TR001433 (Y. Yuan), NIH/NIBIB:R21EB030209 (Y. Yuan), NIH/NCI:R37CA214955 (A. Rao), and NIH:R01CA233888 (A.L. Simpson). The authors would also like to acknowledge the following NIH funded awards for the multi-site clinical trial (NCT00884741, RTOG0825/ACRIN6686): U10CA21661, U10CA37422, U10CA180820, U10CA180794, U01CA176110, R01CA082500, CA079778, CA080098, CA180794, CA180820, CA180822, CA180868. Research reported in this publication was also partly supported by the National Science Foundation, under award numbers 2040532 (S. Baek), and 2040462 (B. Landman). Research reported in this publication was also supported by i) a research grant from Varian Medical Systems (Palo Alto, CA, USA) (Y.Yuan), (ii) the Ministry of Health of the Czech Republic (Grant Nr. NU21-08-00359) (M.Kerkovský and M.Kozubek), (iii) Deutsche Forschungsgemeinschaft (DFG, German Research Foundation) Project-ID 404521405, SFB 1389, Work Package C02, and Priority Program 2177 "Radiomics: Next Generation of Biomedical Imaging" (KI 2410/1-1 | MA 6340/18-1) (P. Vollmuth), (iv) DFG Project-ID B12, SFB 824 (B. Wiestler), (v) the Helmholtz Association (funding number ZT-I-OO1 4) (K. Maier-Hein), vi) the Dutch Cancer Society (KWF project number EMCR 2015-7859) (S.R. van der Voort), (vii) the Chilean National Agency for Research and Development (ANID-Basal FB0008 (AC3E) and FB210017 (CENIA)) (P. Guevara), viii) the Canada CIFAR AI Chairs Program (M. Vallières), (ix) Leeds Hospital Charity (Ref: 9RO1/1403) (S. Currie), (x) the Cancer Research UK funding for the Leeds Radiotherapy Research Centre of Excellence (RadNet) and the grant number C19942/A28832 (S. Currie), (xi) Medical Research Council (MRC) Doctoral Training Program in Precision Medicine (Award Reference No. 2096671) (J. Bernal), (xii) The European Research Council (ERC) under the European Union's Horizon 2020 research and innovation program (Grant Agreement No. 757173) (B.Glocker), (xiii) The UKRI London Medical Imaging & Artificial Intelligence Centre for Value-Based Healthcare (K. Kamnitsas), (xiv) Wellcome/Engineering and Physical Sciences Research Council (EPSRC) Center for Medical Engineering (WT 203148/Z/16/Z) (T.C. Booth), (xv) American Cancer Society Research Scholar Grant RSG-16-005-01 (A. Rao), (xvi) the Department of Defense (DOD)

Peer Reviewed Cancer Research Program (PRCRP) W81XWH-18-1-0404, Dana Foundation David Mahoney Neuroimaging Program, the V Foundation Translational Research Award, Johnson & Johnson WiS-TEM2D Award (P. Tiwari), (xvii) RSNA Research & Education Foundation under grant number RR2011 (E.Calabrese), (xviii) the National Research Fund of Luxembourg (FNR) (grant number: C20/BM/14646004/GLASS-LUX/Niclou) (S.P.Niclou), xix) EU Marie Curie FP7-PEOPLE-2012-ITN project TRANSACT (PITN-GA-2012-316679) and the Swiss National Science Foundation (project number 140958) (J. Slotboom), and (xx) CNPq 303808/2018-7 and FAPESP 2014/12236-1 (A. Xavier Falcão). The content of this publication is solely the responsibility of the authors and does not represent the official views of the NIH, the NSF, the RSNA R&E Foundation, or any of the additional funding bodies.

## Author contributions

Study conception: S. Pati, U. Baid, B. Edwards, M. Sheller, G.A. Reina, J. Martin, S. Bakas. Development of software used in the study: S. Pati, B. Edwards, M. Sheller, S. Wang, G.A. Reina, P. Foley, A. Gruzdev, D. Karkada, S. Bakas. Data acquisition: M. Bilello, S. Mohan, E. Calabrese, J. Rudie, J. Saini, R.Y. Huang, K. Chang, T. So, P. Heng, T.F. Cloughesy, C. Raymond, T. Oughourlian, A. Hagiwara, C. Wang, M. To, M. Kerkovský, T. Koprivová, M. Dostál, V. Vybíhal, J.A. Maldjian, M.C. Pinho, D. Reddy, J. Holcomb, B. Wiestler, M. Metz, R. Jain, M. Lee, P. Tiwari, R. Verma, Y. Gusev, K. Bhuvaneshwar, C. Bencheqroun, A. Belouali, A. Abayazeed, A. Abbassy, S. Gamal, M. Qayati, M. Mekhaimar, M. Reyes, R.R. Colen, M. Ak, P. Vollmuth, G. Brugnara, F. Sahm, M. Bendszus, W. Wick, A. Mahajan, C. Balaña, J. Capellades, J. Puig, Y. Choi, M. Muzi, H.F. Shaykh, A. Herrera-Trujillo, W. Escobar, A. Abello, P. LaMontagne, B. Landman, K. Ramadass, K. Xu, S. Chotai, L.B. Chambless, A. Mistry, R.C. Thompson, J. Bapuraj, N. Wang, S.R. van der Voort, F. Incekara, M.M.J. Wijnenga, R. Gahrmann, J.W. Schouten, H.J. Dubbink, A.J.P.E. Vincent, M.J. van den Bent, H.I. Sair, C.K. Jones, A. Venkataraman, J. Garrett, M. Larson, B. Menze, T. Weiss, M. Weller, A. Bink, Y. Yuan, S. Sharma, T. Tseng, B.C.A. Teixeira, F. Sprenger, S.P. Niclou, O. Keunen, L.V.M. Dixon, M. Williams, R.G.H. Beets-Tan, H. Franco-Maldonado, F. Loayza, J. Slotboom, P. Radojewski, R. Meier, R. Wiest, J. Trenkler, J. Pichler, G. Necker, S. Meckel, E. Torche, F. Vera, E. Lóópez, Y. Kim, H. Ismael, B. Allen, J.M. Buatti, J. Park, P. Zampakis, V. Panagiotopoulos, P. Tsiganos, E. Challiasos, D.M. Kardamakis, P. Prasanna, K.M. Mani, D. Payne, T. Kurc, L. Poisson, M. Vallières, D. Fortin, M. Lepage, F. Morón, J. Mandel, C. Badve, A.E. Sloan, J.S. Barnholtz-Sloan, K. Waite, G. Shukla, S. Liem, G.S. Alexandre, J. Lombardo, J.D. Palmer, A.E. Flanders, A.P. Dicker, G. Ogbole, D. Oyekunle, O. Odafe-Oyibotha, B. Osobu, M. Shu'aibu, F. Dako, A. Dorcas, D. Murcia, R. Haas, J. Thompson, D.R. Ormond, S. Currie, K. Fatania, R. Frood, J. Mitchell, J. Farinhas, A.L. Simpson, J.J. Peoples, R. Hu, D. Cutler, F.Y. Moraes, A. Tran, M. Hamghalam, M.A. Boss, J. Gimpel, B. Bialecki, A. Chelliah. Data processing: C. Sako, S. Ghodasara, E. Calabrese, J. Rudie, M. Jadhav, U. Pandey, R.Y. Huang, M. Jiang, C. Chen, C. Raymond, S. Bhardwaj, C. Chong, M. Agzarian, M. Kozubek, F. Lux, J. Michálek, P. Matula, C. Bangalore Yogananda, D. Reddy, B.C. Wagner, I. Ezhov, M. Lee, Y.W. Lui, R. Verma, R. Bareja, I. Yadav, J. Chen, N. Kumar, K. Bhuvaneshwar, A. Sayah, C. Bencheqroun, K. Kolodziej, M. Hill, M. Reyes, L. Pei, M. Ak, A. Kotrotsou, P. Vollmuth, G. Brugnara, C.J. Preetha, M. Zenk, J. Puig, M. Muzi, H.F. Shaykh, A. Abello, J. Bernal, J. Gómez, P. LaMontagne, K. Ramadass, S. Chotai, N. Wang, M. Smits, S.R. van der Voort, A. Alafandi, F. Incekara, M.M.J. Wijnenga, G. Kapsas, R. Gahrmann, A.J.P.E. Vincent, P.J. French, S. Klein, H.I. Sair, C.K. Jones, J. Garrett, H. Li, F. Kofler, Y. Yuan, S. Adabi, A. Xavier Falcão, S.B. Martins, D. Menotti, D.R. Lucio, O. Keunen, A. Hau, K. Kamnitsas, L. Dixon, S. Benson, E. Pelaez, H. Franco-Maldonado, F. Loayza, S. Quevedo, R. McKinley, J. Trenkler, A. Haunschmidt, C. Mendoza, E. Ríos, J. Choi, S. Baek, J. Yun, P. Zampakis, V. Panagiotopoulos, P. Tsiganos, E.I. Zacharaki, C. Kalogeropoulou, P. Prasanna, S. Shreshtra, T. Kurc, B. Luo, N. Wen, M. Vallières, D. Fortin, F. Morón, C. Badve, V. Vadmal, G. Shukla, G. Ogbole, D. Oyekunle, F. Dako, D. Murcia, E. Fu, S. Currie, R. Frood, M.A. Vogelbaum, J. Mitchell, J. Farinhas, J.J. Peoples, M. Hamghalam, D. Kattil Veettil, K. Schmidt, B. Bialecki, S. Marella, T.C. Booth, A. Chelliah, M. Modat, C. Dragos, H. Shuaib. Data analysis & interpretation: S. Pati, U. Baid, B. Edwards, M. Sheller, S. Bakas. Site PI/Senior member (of each collaborating group): C. Davatzikos, J. Villanueva-Meyer, M. Ingalhalikar, R.Y. Huang, Q. Dou, B.M. Ellingson, M. To, M. Kozubek, J.A. Maldjian, B. Wiestler, R. Jain, P. Tiwari, Y. Gusev, A. Abayazeed, R.R. Colen, P. Vollmuth, A. Mahajan, C. Balaña, S. Lee, M. Muzi, H.F. Shaykh, M. Trujillo, D. Marcus, B. Landman, A. Rao, M. Smits, H.I. Sair, R. Jeraj, B. Menze, Y. Yuan, A. Xavier Falcão, S.P. Niclou, B. Glocker, J. Teuwen, E. Pelaez, R. Wiest, S. Meckel, P. Guevara, S. Baek, H. Kim, D.M. Kardamakis, J. Saltz, L. Poisson, M. Vallières, F. Morón, A.E. Sloan, A.E. Flanders, G. Ogbole, D.R. Ormond, S. Currie, J. Farinhas, A.L. Simpson, C. Apgar, T.C. Booth.Writing the original manuscript: S. Pati, U. Baid, B. Edwards, M. Sheller, S. Bakas. Review, edit, & approval of the final manuscript: All authors.

## Competing interests

The Intel-affiliated authors (B. Edwards, M. Sheller, S. Wang, G.A. Reina, P. Foley, A. Gruzdev, D. Karkada, P. Shah, J. Martin) would like to disclose the following (potential) competing interests as Intel employees. Intel may develop proprietary software that is related in reputation to the OpenFL open source project highlighted in this work. In addition, the work demonstrates feasibility of federated learning for brain tumor boundary detection models. Intel may benefit by selling products to support an increase in demand for this use-case. The remaining authors declare no competing interests.

## Additional information

Sarthak Pati [1,2,3,4,154], Ujjwal Baid [1,2,3,154], Brandon Edwards [5,154], Micah Sheller[5], Shih-Han Wang[5], G. Anthony Reina[5], Patrick Foley [5], Alexey Gruzdev[5], Deepthi Karkada [5], Christos Davatzikos [1,2], Chiharu Sako [1,2], Satyam Ghodasara [2], Michel Bilello[1,2], Suyash Mohan [1,2], Philipp Vollmuth [6], Gianluca Brugnara [6], Chandrakanth J. Preetha [6], Felix Sahm [7,8], Klaus Maier-Hein [9,10], Maximilian Zenk [9], Martin Bendszus[6], Wolfgang Wick [7,11], Evan Calabrese [12], Jeffrey Rudie [12], Javier Villanueva-Meyer[12], Soonmee Cha[12], Madhura Ingalhalikar[13], Manali Jadhav [13], Umang Pandey [13], Jitender Saini[14], John Garrett [15,16], Matthew Larson[15], Robert Jeraj[15,16], Stuart Currie [17], Russell Frood [17], Kavi Fatania [17], Raymond Y. Huang [18], Ken Chang[19], Carmen Balaña [20], Jaume Capellades[21], Josep Puig[22], Johannes Trenkler [23], Josef Pichler[24], Georg Necker [23], Andreas Haunschmidt [23], Stephan Meckel [23,25], Gaurav Shukla[1,26], Spencer Liem[27], Gregory S. Alexander [28], Joseph Lombardo[27,29], Joshua D. Palmer [30], Adam E. Flanders[31], Adam P. Dicker [29], Haris I. Sair [32,33], Craig K. Jones [33], Archana Venkataraman [34], Meirui Jiang [35], Tiffany Y. So [35], Cheng Chen [35], Pheng Ann Heng[35], Qi Dou[35], Michal Kozubek [36], Filip Lux [36], Jan Michálek [36], Petr Matula [36], Miloš Keřkovský [37], Tereza Kopřivová [37], Marek Dostál [37,38], Václav Vybíhal [39], Michael A. Vogelbaum[40], J. Ross Mitchell[41,42], Joaquim Farinhas [43], Joseph A. Maldjian[44], Chandan Ganesh Bangalore Yogananda[44], Marco C. Pinho[44], Divya Reddy[44], James Holcomb[44], Benjamin C. Wagner[44], Benjamin M. Ellingson[45,46], Timothy F. Cloughesy [46], Catalina Raymond[45], Talia Oughourlian[45,47], Akifumi Hagiwara[47], Chencai Wang [47], Minh-Son To[48,49], Sargam Bhardwaj[48], Chee Chong[50], Marc Agzarian [50,51], Alexandre Xavier Falcão [52], Samuel B. Martins [53], Bernardo C. A. Teixeira [54,55], Flávia Sprenger [55], David Menotti [56], Diego R. Lucio[56], Pamela LaMontagne [57], Daniel Marcus[57], Benedikt Wiestler [58,59], Florian Kofler [58,59,60], Ivan Ezhov [4,59,60], Marie Metz [58], Rajan Jain [61,62], Matthew Lee [61], Yvonne W. Lui [61], Richard McKinley [63], Johannes Slotboom [63], Piotr Radojewski[63], Raphael Meier [63], Roland Wiest [63], Derrick Murcia[64], Eric Fu[64], Rourke Haas[64], John Thompson[64], David Ryan Ormond [64], Chaitra Badve [65], Andrew E. Sloan [66,67,68], Vachan Vadmal[68], Kristin Waite [69], Rivka R. Colen [70,71], Linmin Pei[72], Murat Ak [70], Ashok Srinivasan[73], J. Rajiv Bapuraj [73], Arvind Rao[74], Nicholas Wang [74], Ota Yoshiaki[73], Toshio Moritani[73], Sevcan Turk[73], Joonsang Lee [74], Snehal Prabhudesai[74], Fanny Morón [75], Jacob Mandel [51], Konstantinos Kamnitsas [76,77], Ben Glocker [76], Luke V. M. Dixon [78], Matthew Williams [79], Peter Zampakis [80], Vasileios Panagiotopoulos [81], Panagiotis Tsiganos [82], Sotiris Alexiou[83], Ilias Haliassos [84], Evangelia I. Zacharaki [83], Konstantinos Moustakas [83], Christina Kalogeropoulou [80], Dimitrios M. Kardamakis[85], Yoon Seong Choi [86], Seung-Koo Lee [86], Jong Hee Chang [86], Sung Soo Ahn [86], Bing Luo[87], Laila Poisson [88], Ning Wen [87,89], Pallavi Tiwari[90], Ruchika Verma[42,90], Rohan Bareja[90], Ipsa Yadav[90], Jonathan Chen [90], Neeraj Kumar [41,42], Marion Smits [91], Sebastian R. van der Voort[91], Ahmed Alafandi[91], Fatih Incekara[91,92], Maarten M. J. Wijnenga[93], Georgios Kapsas [91], Renske Gahrmann [91], Joost W. Schouten[92], Hendrikus J. Dubbink [94], Arnaud J. P. E. Vincent [92], Martin J. van den Bent [93], Pim J. French [93], Stefan Klein [95], Yading Yuan [96], Sonam Sharma[96], Tzu-Chi Tseng[96], Saba Adabi[96], Simone P. Niclou [97], Olivier Keunen [98], Ann-Christin Hau [97,99], Martin Vallières [100,101], David Fortin[101,102], Martin Lepage [101,103], Bennett Landman [104], Karthik Ramadass[104], Kaiwen Xu [105], Silky Chotai[106], Lola B. Chambless[106], Akshitkumar Mistry[106], Reid C. Thompson[106], Yuriy Gusev [107], Krithika Bhuvaneshwar [107], Anousheh Sayah [108], Camelia Bencheqroun[107], Anas Belouali [107], Subha Madhavan[107], Thomas C. Booth [109,110], Alysha Chelliah[109], Marc Modat[109], Haris Shuaib [111,112], Carmen Dragos [111], Aly Abayazeed[113], Kenneth Kolodziej[113], Michael Hill[113], Ahmed Abbassy[114], Shady Gamal[114], Mahmoud Mekhaimar[114], Mohamed Qayati [114], Mauricio Reyes [115], Ji Eun Park[116], Jihye Yun[116], Ho Sung Kim [116], Abhishek Mahajan [117], Mark Muzi [118], Sean Benson [119], Regina G. H. Beets-Tan[120,121], Jonas Teuwen[119], Alejandro Herrera-Trujillo [122,123], Maria Trujillo[123], William Escobar[122,123], Ana Abello[123], Jose Bernal [123,124], Jhon Gómez[123], Joseph Choi[125], Stephen Baek [126], Yusung Kim[127], Heba Ismael[127], Bryan Allen [127], John M. Buatti[127], Aikaterini Kotrotsou[128], Hongwei Li[129], Tobias Weiss [130], Michael Weller [130], Andrea Bink [131], Bertrand Pouymayou[131], Hassan F. Shaykh[132], Joel Saltz [133], Prateek Prasanna[133], Sampurna Shrestha [133], Kartik M. Mani [133,134], David Payne[135], Tahsin Kurc [133,136], Enrique Pelaez[137], Heydy Franco-Maldonado[138], Francis Loayza[137], Sebastian Quevedo [139], Pamela Guevara [140], Esteban Torche[140], Cristobal Mendoza[140], Franco Vera[140], Elvis Ríos[140], Eduardo López[140], Sergio A. Velastin [141], Godwin Ogbole [142], Mayowa Soneye[142], Dotun Oyekunle [142], Olubunmi Odafe-Oyibotha[143], Babatunde Osobu[142], Mustapha Shu'aibu[144], Adeleye Dorcas[145], Farouk Dako [2,146], Amber L. Simpson[112,147], Mohammad Hamghalam[147,148], Jacob J. Peoples [147], Ricky Hu[147], Anh Tran [147], Danielle Cutler[149], Fabio Y. Moraes [150], Michael A. Boss [151], James Gimpel [151], Deepak Kattil Veettil [151], Kendall Schmidt[152], Brian Bialecki [152], Sailaja Marella[151], Cynthia Price[151], Lisa Cimino[151], Charles Apgar[151], Prashant Shah [5], Bjoern Menze [4,129], Jill S. Barnholtz-Sloan [69,153], Jason Martin [5] & Spyridon Bakas [1,2,3] ✉

[1]Center for Biomedical Image Computing and Analytics (CBICA), University of Pennsylvania, Philadelphia, PA, USA. [2]Department of Radiology, Perelman School of Medicine, University of Pennsylvania, Philadelphia, PA, USA. [3]Department of Pathology and Laboratory Medicine, Perelman School of Medicine, University of Pennsylvania, Philadelphia, PA, USA. [4]Department of Informatics, Technical University of Munich, Munich, Bavaria, Germany. [5]Intel Corporation, Santa Clara, CA, USA. [6]Department of Neuroradiology, Heidelberg University Hospital, Heidelberg, Germany. [7]Clinical Cooperation Unit Neuropathology, German Cancer Consortium (DKTK) within the German Cancer Research Center (DKFZ), Heidelberg, Germany. [8]Department of Neuropathology, Heidelberg University Hospital, Heidelberg, Germany. [9]Division of Medical Image Computing, German Cancer Research Center, Heidelberg, Germany. [10]Pattern Analysis and Learning Group, Department of Radiation Oncology, Heidelberg University Hospital, Heidelberg, Germany. [11]Neurology Clinic, Heidelberg University Hospital, Heidelberg, Germany. [12]Department of Radiology & Biomedical Imaging, University of California San Francisco, San Francisco, CA, USA. [13]Symbiosis Center for Medical Image Analysis, Symbiosis International University, Pune, Maharashtra, India. [14]Department of Neuroimaging and Interventional Radiology, National Institute of Mental Health and Neurosciences, Bangalore, Karnataka, India. [15]Department of Radiology, School of Medicine and Public Health, University of Wisconsin, Madison, WI, USA. [16]Department of Medical Physics, School of Medicine and Public Health, University of Wisconsin, Madison, WI, USA. [17]Leeds Teaching Hospitals Trust, Department of Radiology, Leeds, UK. [18]Department of Radiology, Brigham and Women's Hospital, Harvard Medical School, Boston, MA, USA. [19]Athinoula A. Martinos Center for Biomedical Imaging, Massachusetts General Hospital, Charlestown, MA, USA. [20]Catalan Institute of Oncology, Badalona, Spain. [21]Consorci MAR Parc de Salut de Barcelona, Catalonia, Spain. [22]Department of Radiology (IDI), Girona Biomedical Research Institute (IdIBGi), Josep Trueta University Hospital, Girona, Spain. [23]Institute of Neuroradiology, Neuromed Campus (NMC), Kepler University Hospital Linz, Linz, Austria. [24]Department of Neurooncology, Neuromed Campus (NMC), Kepler University Hospital Linz, Linz, Austria. [25]Institute of Diagnostic and Interventional Neuroradiology, RKH Klinikum Ludwigsburg, Ludwigsburg, Germany. [26]Department of Radiation Oncology, Christiana Care Health System, Philadelphia, PA, USA. [27]Sidney Kimmel Medical College, Thomas Jefferson University, Philadelphia, PA, USA. [28]Department of Radiation Oncology, University of Maryland, Baltimore, MD, USA. [29]Department of Radiation Oncology, Sidney Kimmel Cancer Center, Thomas Jefferson University, Philadelphia, PA, USA. [30]Department of Radiation Oncology, The James Cancer Hospital and Solove Research Institute, The Ohio State University Comprehensive Cancer Center, Columbus, OH, USA. [31]Department of Radiology, Sidney Kimmel Cancer Center, Thomas Jefferson University, Philadelphia, PA, USA. [32]The Russell H. Morgan Department of Radiology and Radiological Science, Johns Hopkins University School of Medicine, Baltimore, MD, USA. [33]The Malone Center for Engineering in Healthcare, The Whiting School of Engineering, Johns Hopkins University, Baltimore, MD, USA. [34]Department of Electrical and Computer Engineering, Whiting School of Engineering, Johns Hopkins University, Baltimore, MD, USA. [35]The Chinese University of Hong Kong, Hong Kong, China. [36]Centre for Biomedical Image Analysis, Faculty of Informatics, Masaryk University, Brno, Czech Republic. [37]Department of Radiology and Nuclear Medicine, Faculty of Medicine, Masaryk University, Brno and University Hospital Brno, Brno, Czech Republic. [38]Department of Biophysics, Faculty of Medicine, Masaryk University, Brno, Czech Republic. [39]Department of Neurosurgery, Faculty of Medicine, Masaryk University, Brno, and University Hospital and Czech Republic, Brno, Czech Republic. [40]Department of Neuro Oncology, H. Lee Moffitt Cancer Center and Research Institute, Tampa, FL, USA. [41]University of Alberta, Edmonton, AB, Canada. [42]Alberta Machine Intelligence Institute, Edmonton, AB, Canada. [43]Department of Radiology, H. Lee Moffitt Cancer Center and Research Institute, Tampa, FL, USA. [44]University of Texas Southwestern Medical Center, Dallas, TX, USA. [45]UCLA Brain Tumor Imaging Laboratory (BTIL), Center for Computer Vision and Imaging Biomarkers, Department of Radiological Sciences, David Geffen School of Medicine, University of California Los Angeles, Los Angeles, CA, USA. [46]UCLA Neuro-Oncology Program, Department of Neurology, David Geffen School of Medicine, University of California Los Angeles, Los Angeles, CaA, USA. [47]Department of Radiological Sciences, David Geffen School of Medicine, University of California Los Angeles, Los Angeles, CA, USA. [48]College of Medicine and Public Health, Flinders University, Bedford Park, SA, Australia. [49]Division of Surgery and Perioperative Medicine, Flinders Medical Centre, Bedford Park, SA, Australia. [50]South Australia Medical Imaging, Flinders Medical Centre, Bedford Park, SA, Australia. [51]Department of Neurology, Baylor College of Medicine, Houston, TX, USA. [52]Institute of Computing, University of Campinas, Campinas, São Paulo, Brazil. [53]Federal Institute of São Paulo, Campinas, São Paulo, Brazil. [54]Instituto de Neurologia de Curitiba, Curitiba, Paraná, Brazil. [55]Department of Radiology, Hospital de Clínicas da Universidade Federal do Paraná, Curitiba, Paraná, Brazil. [56]Department of Informatics, Universidade Federal do Paraná, Curitiba, Paraná, Brazil. [57]Department of Radiology, Washington University in St. Louis, St. Louis, MO, USA. [58]Department of Diagnostic and Interventional Neuroradiology, School of Medicine, Klinikum rechts der Isar, Technical University of Munich, Munich, Germany. [59]TranslaTUM (Zentralinstitut für translationale Krebsforschung der Technischen Universität München), Klinikum rechts der Isar, Munich, Germany. [60]Image-Based Biomedical Modeling, Department of Informatics, Technical University of Munich, Munich, Germany. [61]Department of Radiology, NYU Grossman School of Medicine, New York, NY, USA. [62]Department of Neurosurgery, NYU Grossman School of Medicine, New York, NY, USA. [63]Support Center for Advanced Neuroimaging, University Institute of Diagnostic and Interventional Neuroradiology, University Hospital Bern, Inselspital, University of Bern, Bern, Switzerland. [64]Department of Neurosurgery, Anschutz Medical Campus, University of Colorado, Aurora, CO, USA. [65]Department of Radiology, University Hospitals Cleveland, Cleveland, OH, USA. [66]Department of Neurological Surgery, University Hospitals-Seidman Cancer Center, Cleveland, OH, USA. [67]Case Comprehensive Cancer Center, Cleveland, OH, USA. [68]Department of Neurosurgery, Case Western Reserve University School of Medicine, Cleveland, OH, USA. [69]National Cancer Institute, National Institute of Health, Division of Cancer Epidemiology and Genetics, Bethesda, MD, USA. [70]Department of Radiology, Neuroradiology Division, University of Pittsburgh, Pittsburgh, PA, USA. [71]Department of Diagnostic Radiology, University of Texas MD Anderson Cancer Center, Houston, TX, USA. [72]University of Pittsburgh Medical Center, Pittsburgh, PA, USA. [73]Department of Neuroradiology, University of Michigan, Ann Arbor, MI, USA. [74]Department of Computational Medicine and Bioinformatics, University of Michigan, Ann Arbor, MI, USA. [75]Department of Radiology, Baylor College of Medicine, Houston, TX, USA. [76]Department of Computing, Imperial College London, London, UK. [77]Institute of Biomedical Engineering, Department of Engineering Science, University of Oxford, Oxford, UK. [78]Department of Radiology, Imperial College NHS Healthcare Trust, London, UK. [79]Computational Oncology Group, Institute for Global Health Innovation, Imperial College London, London, UK. [80]Department of NeuroRadiology, University of Patras, Patras, Greece. [81]Department of Neurosurgery, University of Patras, Patras, Greece. [82]Clinical Radiology Laboratory, Department of Medicine, University of Patras, Patras, Greece. [83]Department of Electrical and Computer Engineering, University of Patras, Patras, Greece. [84]Department of Neuro-Oncology, University of Patras, Patras, Greece. [85]Department of Radiation Oncology, University of Patras, Patras, Greece. [86]Yonsei University College of Medicine, Seoul, Korea. [87]Department of Radiation Oncology, Henry Ford Health System, Detroit, MI, USA. [88]Public Health Sciences, Henry Ford Health System, Detroit, MI, USA. [89]SJTU-Ruijin-UIH Institute for Medical Imaging Technology, Ruijin Hospital, Shanghai Jiao Tong University School of Medicine, 200025 Shanghai, China. [90]Case Western Reserve University, Cleveland, OH, USA. [91]Department of Radiology and Nuclear Medicine, Erasmus MC University Medical Centre Rotterdam, Rotterdam, Netherlands. [92]Department of Neurosurgery, Brain Tumor Center, Erasmus MC University Medical Centre Rotterdam, Rotterdam, Netherlands. [93]Department of Neurology, Brain Tumor Center, Erasmus MC Cancer Institute, Rotterdam, Netherlands. [94]Department of Pathology, Brain Tumor Center, Erasmus MC Cancer Institute, Rotterdam, Netherlands. [95]Biomedical Imaging Group Rotterdam, Department of Radiology and Nuclear Medicine, Erasmus MC University Medical Centre Rotterdam, Rotterdam, Netherlands. [96]Department of Radiation Oncology, Icahn School of Medicine at Mount Sinai, New York, NY, USA. [97]NORLUX Neuro-

Oncology Laboratory, Department of Cancer Research, Luxembourg Institute of Health, Luxembourg, Luxembourg. [98]Translation Radiomics, Department of Cancer Research, Luxembourg Institute of Health, Luxembourg, Luxembourg. [99]Luxembourg Center of Neuropathology, Laboratoire National De Santé, Luxembourg, Luxembourg. [100]Department of Computer Science, Université de Sherbrooke, Sherbrooke, QC, Canada. [101]Centre de Recherche du Centre Hospitalière Universitaire de Sherbrooke, Sherbrooke, QC, Canada. [102]Division of Neurosurgery and Neuro-Oncology, Faculty of Medicine and Health Science, Université de Sherbrooke, Sherbrooke, QC, Canada. [103]Department of Nuclear Medicine and Radiobiology, Sherbrooke Molecular Imaging Centre, Université de Sherbrooke, Sherbrooke, QC, Canada. [104]Electrical and Computer Engineering, Vanderbilt University, Nashville, TN, USA. [105]Department of Computer Science, Vanderbilt University, Nashville, TN, USA. [106]Department of Neurosurgery, Vanderbilt University Medical Center, Nashville, TN, USA. [107]Innovation Center for Biomedical Informatics (ICBI), Georgetown University, Washington, DC, USA. [108]Division of Neuroradiology & Neurointerventional Radiology, Department of Radiology, MedStar Georgetown University Hospital, Washington, DC, USA. [109]School of Biomedical Engineering & Imaging Sciences, King's College London, London, UK. [110]Department of Neuroradiology, Ruskin Wing, King's College Hospital NHS Foundation Trust, London, UK. [111]Stoke Mandeville Hospital, Mandeville Road, Aylesbury, UK. [112]Department of Biomedical and Molecular Sciences, Queen's University, Kingston, ON, Canada. [113]Neosoma Inc., Groton, MA, USA. [114]University of Cairo School of Medicine, Giza, Egypt. [115]University of Bern, Bern, Switzerland. [116]Department of Radiology, Asan Medical Center, Seoul, South Korea. [117]The Clatterbridge Cancer Centre NHS Foundation Trust Pembroke Place, Liverpool, UK. [118]Department of Radiology, University of Washington, Seattle, WA, USA. [119]Netherlands Cancer Institute, Amsterdam, Netherlands. [120]Department of Radiology, Netherlands Cancer Institute, Amsterdam, Netherlands. [121]GROW School of Oncology and Developmental Biology, Maastricht, Netherlands. [122]Clínica Imbanaco Grupo Quirón Salud, Cali, Colombia. [123]Universidad del Valle, Cali, Colombia. [124]The University of Edinburgh, Edinburgh, UK. [125]Department of Industrial and Systems Engineering, University of Iowa, Iowa, USA. [126]Department of Industrial and Systems Engineering, Department of Radiation Oncology, University of Iowa, Iowa City, IA, USA. [127]Department of Radiation Oncology, University of Iowa, Iowa City, IA, USA. [128]MD Anderson Cancer Center, University of Texas, Houston, TX, USA. [129]Department of Quantitative Biomedicine, University of Zurich, Zurich, Switzerland. [130]Department of Neurology, Clinical Neuroscience Center, University Hospital Zurich and University of Zurich, Zurich, Switzerland. [131]Department of Neuroradiology, Clinical Neuroscience Center, University Hospital Zurich and University of Zurich, Zurich, Switzerland. [132]University of Alabama in Birmingham, Birmingham, AL, USA. [133]Department of Biomedical Informatics, Stony Brook University, Stony Brook, New York, USA. [134]Department of Radiation Oncology, Stony Brook University, Stony Brook, NY, USA. [135]Department of Radiology, Stony Brook University, Stony Brook, NY, USA. [136]Scientific Data Group, Oak Ridge National Laboratory, Oak Ridge, TN, USA. [137]Escuela Superior Politecnica del Litoral, Guayaquil, Guayas, Ecuador. [138]Sociedad de Lucha Contral el Cancer - SOLCA, Guayaquil Ecuador, Guayaquil, Ecuador. [139]Universidad Católica de Cuenca, Cuenca, Ecuador. [140]Universidad de Concepción, Concepción, Biobío, Chile. [141]School of Electronic Engineering and Computer Science, Queen Mary University of London, London, UK. [142]Department of Radiology, University College Hospital Ibadan, Oyo, Nigeria. [143]Clinix Healthcare, Lagos, Lagos, Nigeria. [144]Department of Radiology, Muhammad Abdullahi Wase Teaching Hospital, Kano, Nigeria. [145]Department of Radiology, Obafemi Awolowo University Ile-Ife, Ile-Ife, Osun, Nigeria. [146]Center for Global Health, Perelman School of Medicine, University of Pennsylvania, Philadelphia, PA, USA. [147]School of Computing, Queen's University, Kingston, ON, Canada. [148]Department of Electrical Engineering, Qazvin Branch, Islamic Azad University, Qazvin, Iran. [149]The Faculty of Arts & Sciences, Queen's University, Kingston, ON, Canada. [150]Department of Oncology, Queen's University, Kingston, ON, Canada. [151]Center for Research and Innovation, American College of Radiology, Philadelphia, PA, USA. [152]Data Science Institute, American College of Radiology, Reston, VA, USA. [153]Center for Biomedical Informatics and Information Technology, National Cancer Institute (NCI), National Institute of Health, Bethesda, MD, USA. [154]These authors contributed equally: Sarthak Pati, Ujjwal Baid, Brandon Edwards. ✉e-mail: sbakas@upenn.edu

