## [Peer Review File · Nature Communications]

Federated Learning Enables Big Data for Rare Cancer Boundary DetectionEditorial Note: Parts of this Peer Review File have been redacted as indicated to maintain the confidentiality of unpublished data.

Reviewers' Comments:

Reviewer #2:

Remarks to the Author:

In this work, the authors present a large-scale Federated Learning approach for detecting glioblastoma sub-compartment boundaries. The work focuses primarily on the unique obstacles which must be overcome in expansive collaborative settings, as well as the myriad benefits that federated learning can provide to the medical community. The authors clearly demonstrate federated learning's increased model performance, robustness to data quality issues, and affinity for complex tasks.

Of particular interest was the discussion in which the authors mention the administrative steps taken to ensure smooth collaboration. In fact, I think this discussion may be of more interest to the research community than the empirical results demonstrated in this work, promising though they may be. One of the largest obstacles to federated learning in medicine is creating a workflow which is streamlined for the human researchers involved in the federated consortium, and the steps the authors took to coordinate the scale and diversity of data-collection sites are all reasonable and impressive.

Toward the end of the work, the author's claim that their efforts can "serve as a blueprint for future FL studies that result in clinically deployable ML models". Although I think this study is a fine demonstration of federated learning, and it provides many important insights, I think this claim is overzealous. The provided work does not sufficiently demonstrate methods that would generalize to other studies. I will now raise some specific concerns here which speak to the limitations in this work.

The methods, from a machine learning and federated learning standpoint, are all straightforward. Only one strategy for federated learning is utilized (a single, trusted server which serves as aggregator) the decision to utilize this strategy is unjustified in the work. Although there are clear situations where a single aggregator model makes perfect sense, and is in fact the optimal choice, many other federated learning strategies exist in the literature, and are useful in other situations. Single aggregator settings may not be desirable, for example, in settings where the trusted aggregator is distantly connected to some members of the consortium - multiple aggregators at different locations may be the better choice in this situation, especially if runtime is of concern. To be clear, I do not think that the authors made the incorrect choice by using the single aggregator paradigm; however, this is one example how the framework may not generalize outside of the choices the authors made for their own work.

Another significant gap is the lack of focus on data privacy in this work. To their credit, the authors do mention that federated learning both can provide solutions to privacy problems (don't have to transfer data between sites) and can suffer from them (difficult to debug if private data causes problems) - these are both correct and important insights. Beyond this however, the only mention of security measures taken for the study is that collaborators "securely connected" to a central aggregation server, and that the aggregation server is "trusted". In order to support these claims, more details are needed to establish what kinds of measures were taken.

Even if more details were provided, a secure connection and trusted aggregators do not ensure data privacy, especially when model parameters are passed without perturbation between collaborators. Some fully-trained deep-learning models, for example, can be used to identify the participation of individuals in the data set (see for particular examples [1, 2] and [3,4] for nice surveys).

I firmly believe that these issues of privacy are vitally important to consider in federated learning, and many efforts in the research community have been made in the community to address them (e.g. applications of differential privacy [4], many). I do think it is a detriment to this work that no more details on privacy-preserving efforts are given, and I do think inclusion of some more detail on that front is necessary. I also believe that this further demonstrates that this work does not serve as a blueprint (privacy issues are especially nuanced!), and would like to reiterate my request that the authors tone down that claim. A sufficient blueprint would also need to do more

work to address issues in federated learning such as stability to collaborator dropout, bottlenecks in communication between sites, financial considerations, and much more which this work does not.

To move away from critical discussion - I do think the scope of the experiments and the work done are strong, and the study stands on its own as a particularly interesting application. The scope and diversity of the study is impressive, and the potential benefits to the research community are clear. I think, though, that this work would benefit more from taking pride in being a wonderfully fascinating case-study, and a marvelous exercise in large-scale federated learning, rather than a generic blueprint which future studies ought to follow - such a claim is simply beyond the scope of this otherwise strong work. In future work, I would encourage the authors to dig more into why certain choices are made during the federated methodology, and to justify why those decisions were made clear. I would also like to see more detailed discussion on how the administrative tasks were handled - I do firmly believe these insights would significantly benefit the community as a whole by providing a dimension of collaboration which is often omitted from technical papers.

—

[1] Lam, Maximilian, et al. "Gradient disaggregation: Breaking privacy in federated learning by reconstructing the user participant matrix." International Conference on Machine Learning. PMLR, 2021

[2] Oh, Hyunseok, and Youngki Lee. "Exploring image reconstruction attack in deep learning computation offloading." The 3rd International Workshop on Deep Learning for Mobile Systems and Applications. 2019.

[3] Hu, Hongsheng, et al. "Membership inference attacks on machine learning: A survey." ACM Computing Surveys (CSUR) (2021).

[4] Liu, Ximeng, et al. "Privacy and security issues in deep learning: A survey." IEEE Access 9 (2020): 4566-4593.

[5] Dwork, Cynthia. "Differential privacy: A survey of results." International conference on theory and applications of models of computation. Springer, Berlin, Heidelberg, 2008.

Reviewer #3:

Remarks to the Author:

In this study titled "Federated Learning Enables Big Data for Rare Cancer Boundary Detection," the authors use FML from 71 healthcare institutions across 6 continents to generate an automatic tumor boundary detector for glioblastoma. They report a 33% increased improvement over a publicly trained model to delineate surgical targetable tumor. This is a very comprehensive study with a large amount of data showing the utility of FML.

1. Why was glioblastoma chosen? The rarity is not as rare as other such diseases so not sure if the FML will be as applicable as to say something like an endolymphatic sac tumor, or even low grade glioma for that matter.
2. How was the publicly trained model developed? Why not compare to established ML models?
3. What about distinguishing and identifying non enhancing glioblastoma?
4. Where did the FML have the most difficulty?

Reviewer #4:

Remarks to the Author:

There is no doubt that such a global project in medical imaging will enrich our understanding of the benefits of AI for patient care. Thank you to everybody involved.

Below is a list of my concerns in no particular order:

- DSC is known to deliver higher numbers than IoU (Jaccard index). For a more rigorous evaluation, it would have been much more valuable to report both numbers.
- There is virtually no information on the actual federated learning used. One can only guess that the authors have used the original federated averaging (FedAvg). More details on the federated mechanism would be helpful.
- The six sites selected for the out-of-sample generalization test (which is the main motivation for using FL) have been selected due to availability. This is rather a point of concern. Random selection would have been more reliable.
- Image samples have not been provided to show the difference between the public initial model and the final consensus model for small, moderate and large differences.
- The data quality issue regarding erroneous reference annotations, or inherent variability, has not been properly addressed, among others, for the validation through six sites. Were the 590 cases delineated by one expert only?
- A more in-depth and conclusive analysis regarding the comparison of the ensemble approach and the FL network is missing. Would a network of few major hospitals have the same effect as a complicated global network?
- Why were you not able to evaluate the exclusion of problematic sites like 48?
- Recent literature has not been mentioned. For instance:
 - 1) ProxyFL: Decentralized Federated Learning through Proxy Model Sharing
 - 2) Federated Learning with Differential Privacy: Algorithms and Performance Analysis
 - 3) Federated learning and differential privacy for medical image analysis
 - 4) Federated Learning for Computational Pathology on Gigapixel Whole Slide Images

Response to reviewers document

Manuscript Title: Federated Learning Enables Big Data for Rare Cancer Boundary Detection

Submitted Journal: Nature Communications

We thank all reviewers for providing their feedback and thoughtful comments on our manuscript. In the revised version, we have duly incorporated the feedback provided by the editor and the reviewers to further improve the quality of our manuscript.

We specifically thank **Reviewer #2** for recognizing our attempt to “clearly demonstrate federated learning’s increased model performance, robustness to data quality issues, and affinity for complex tasks”, for finding “Of particular interest was the discussion in which the authors mention the administrative steps taken to ensure smooth collaboration”, and for characterizing “the steps the authors took to coordinate the scale and diversity of data-collection sites are all reasonable and impressive.” We also want to thank **Reviewer #3** for finding our manuscript “a very comprehensive study with a large amount of data showing the utility of FML.”, and **Reviewer #4** for expressing that “There is no doubt that such a global project in medical imaging will enrich our understanding of the benefits of AI for patient care. Thank you to everybody involved.”

We further want to thank the editorial team for appreciating the quality of our manuscript and offering us the opportunity to submit the revised version of our manuscript.

Below we have listed a point-by-point response to all the reviewers’ comments, including verbatim their original remarks. The reviewers’ comments are reproduced in blue font followed by our responses in black font. Additionally, all associated changes in the manuscript are indicated by different font color, where new additions are indicated in blue font color and any deletions in red.

REVIEWER #2

In this work, the authors present a large-scale Federated Learning approach for detecting glioblastoma sub-compartment boundaries. The work focuses primarily on the unique obstacles which must be overcome in expansive collaborative settings, as well as the myriad benefits that federated learning can provide to the medical community. The authors clearly demonstrate federated learning’s increased model performance, robustness to data quality issues, and affinity for complex tasks.

Of particular interest was the discussion in which the authors mention the administrative steps taken to ensure smooth collaboration. In fact, I think this discussion may be of more interest to the research community than the empirical results demonstrated in this work, promising though they may be. One of the largest obstacles to federated learning in medicine is creating a workflow which is streamlined for the human researchers involved in the federated consortium, and the steps the authors took to coordinate the scale and diversity of data-collection sites are all reasonable and impressive.

RESPONSE:

We thank the reviewer for their encouraging comments and the insights they garnered through our work. We agree with the reviewer’s comment on one of the largest obstacles being the streamlined workflow and the coordination of the numerous institutions. We have indeed experienced during our

study that the administrative component of the study was the most time-consuming, particularly considering the scale of the study. To this end, and to highlight this point further, we have now made a further addition in section Discussion (page 13, line 521) of the manuscript (shown below for the convenience of the reviewer as underlined text and including some preceding text to put the new addition in context)

Current excerpt (with additions underlined): ... "By ``governance" of the federation we refer both to the accurate definition of the problem statement (including reference labels and harmonization considerations accounting for inter-site variability), and the coordination with the collaborating sites for eligibility and compliance with the problem statement definition, as well as security and technical considerations. For future efforts aiming to conduct studies of similar global scale, it would be beneficial to identify a solution for governance prior to initiating the study itself.

Toward the end of the work, the author' claim that their efforts can "serve as a blueprint for future FL studies that result in clinically deployable ML models". Although I think this study is a fine demonstration of federated learning, and it provides many important insights, I think this claim is overzealous. The provided work does not sufficiently demonstrate methods that would generalize to other studies.

RESPONSE:

In retrospect, we agree with the reviewer and we have now toned down this claim and removed any reference to the study serving as a blueprint.

I will now raise some specific concerns here which speak to the limitations in this work.

The methods, from a machine learning and federated learning standpoint, are all straightforward. Only one strategy for federated learning is utilized (a single, trusted server which serves as aggregator) the decision to utilize this strategy is unjustified in the work. Although there are clear situations where a single aggregator model makes perfect sense, and is in fact the optimal choice, many other federated learning strategies exist in the literature, and are useful in other situations. Single aggregator settings may not be desirable, for example, in settings where the trusted aggregator is distantly connected to some members of the consortium - multiple aggregators at different locations may be the better choice in this situation, especially if runtime is of concern. To be clear, I do not think that the authors made the incorrect choice by using the single aggregator paradigm; however, this is one example how the framework may not generalize outside of the choices the authors made for their own work.

RESPONSE:

We thank the reviewer for this observation, and the generally constructive angle of the provided remarks. We acknowledge that various FL workflows/strategies would need to be evaluated for different problem types, and we understand that we have not sufficiently emphasized that this is the process we followed too. We have now made further edits in the existing related text we had in the introduction (page 6, line 311) and in the methods section (page 19, line 790) of our manuscript, in our attempt to further emphasize the fact that we have indeed undergone such a thorough performance evaluation (published in Sci Rep [2]) for this very same task (i.e., detecting glioblastoma sub-compartment boundaries), in order to identify the optimal workflow. The specific excerpt from the introduction section is included below for the reviewer's convenience:

Current excerpt (with additions underlined): "Co-authors in this study have previously introduced FL in healthcare in a simulated setting [1] and further performed a thorough quantitative performance evaluation of different FL workflows [2] (Fig.4) for the same use-case as the present study, i.e., detecting the boundaries of glioblastoma sub-compartments. Findings from these studies [1,2] supported the superiority of the FL workflow used in the present study (i.e., based on an aggregation server [3, 4]) that had almost identical performance to CL, for this use-case."

Newly added text (Methods section (page 19, line 790)): We have identified this FL workflow (based on a central aggregation server) as the optimal for this use-case, following a performance evaluation [2] for this very same task, i.e., detecting glioblastoma sub-compartment boundaries.

Further acknowledging that other forms of model aggregation may be beneficial given compute and network constraints, we have also made the following additions in the Methods section (page 19, line 799):

Current excerpt (with additions underlined): ... Each such iteration is called a "federated round". Based on our previously conducted performance evaluation for this use-case [2], we chose to perform aggregation of all collaborator updates in the present study, using the federated averaging (FedAvg) approach [3], i.e., average of collaborator's model updates weighted according to collaborator's contributing data. We expect these aggregation strategy choices to be use-case dependent, by providing due consideration to the collaborators' associated compute and network infrastructure.

- [1] Sheller, Micah J., et al. "Multi-institutional deep learning modeling without sharing patient data: A feasibility study on brain tumor segmentation." International MICCAI Brainlesion Workshop. Springer, Cham, 2018.
- [2] Sheller, Micah J., et al. "Federated learning in medicine: facilitating multi-institutional collaborations without sharing patient data." Scientific reports 10.1 (2020): 1-12.
- [3] McMahan, Brendan, et al. "Communication-efficient learning of deep networks from decentralized data." Artificial intelligence and statistics. PMLR, 2017.
- [4] Rieke, Nicola, et al. "The future of digital health with federated learning." NPJ digital medicine 3.1 (2020): 1-7.

Another significant gap is the lack of focus on data privacy in this work. To their credit, the authors do mention that federated learning both can provide solutions to privacy problems (don't have to transfer data between sites) and can suffer from them (difficult to debug if private data causes problems) - these are both correct and important insights. Beyond this however, the only mention of security measures taken for the study is that collaborators "securely connected" to a central aggregation server, and that the aggregation server is "trusted". In order to support these claims, more details are needed to establish what kinds of measures were taken.

RESPONSE:

We thank the reviewer for this thoughtful observation. We have now added appropriate text in the Methods section (page 19, line 804) of our manuscript, to further clarify what technologies were used

in our study towards increasing privacy and security. This text follows the description of using federated averaging added to address the previous comment and reads as follows:

Newly added text: ... *In this study, all the network communications during the FL model training process were based on Transport Layer Security (TLS) [4], to mitigate potential exposure of information during transit. Additionally, we demonstrated the feasibility of Trusted Execution Environments (TEEs) [5, 6] for federated training by running the aggregator workload on the secure enclaves of Intel's Secure Guard Extensions (SGX) hardware (Intel® Xeon® E-2286M vPro 8-Core 2.4-5.0GHz Turbo), which ensured the confidentiality of the updates being aggregated and the integrity of the consensus model. TLS and TEEs can help mitigate some of the security and privacy concerns that remain for FL [7].*

- [1] Sheller, Micah J., et al. "Federated learning in medicine: facilitating multi-institutional collaborations without sharing patient data." *Scientific reports* 10.1 (2020): 1-12.
- [2] McMahan, Brendan, et al. "Communication-efficient learning of deep networks from decentralized data." *Artificial intelligence and statistics*. PMLR, 2017.
- [3] Warnat-Herresthal, Stefanie, et al. "Swarm learning for decentralized and confidential clinical machine learning." *Nature* 594.7862 (2021): 265-270.
- [4] Knauth, Thomas, et al. "Integrating remote attestation with transport layer security." *arXiv preprint arXiv:1801.05863* (2018).
- [5] Sabt, Mohamed, Mohammed Achemlal, and Abdelmadjid Bouabdallah. "Trusted execution environment: what it is, and what it is not." *2015 IEEE Trustcom/BigDataSE/ISPA*. Vol. 1. IEEE, 2015.
- [6] Schneider, Moritz, et al. "SoK: Hardware-supported Trusted Execution Environments." *arXiv preprint arXiv:2205.12742* (2022).
- [7] Kairouz, Peter, et al. "Advances and open problems in federated learning." *Foundations and Trends® in Machine Learning* 14.1–2 (2021): 1-210.

Even if more details were provided, a secure connection and trusted aggregators do not ensure data privacy, especially when model parameters are passed without perturbation between collaborators. Some fully-trained deep-learning models, for example, can be used to identify the participation of individuals in the data set (see for particular examples [1, 2] and [3,4] for nice surveys).

- [1] Lam, Maximilian, et al. "Gradient disaggregation: Breaking privacy in federated learning by reconstructing the user participant matrix." *International Conference on Machine Learning*. PMLR, 2021
- [2] Oh, Hyunseok, and Youngki Lee. "Exploring image reconstruction attack in deep learning computation offloading." *The 3rd International Workshop on Deep Learning for Mobile Systems and Applications*. 2019.
- [3] Hu, Hongsheng, et al. "Membership inference attacks on machine learning: A survey." *ACM Computing Surveys (CSUR)* (2021).
- [4] Liu, Ximeng, et al. "Privacy and security issues in deep learning: A survey." *IEEE Access* 9 (2020): 4566-4593.
- [5] Dwork, Cynthia. "Differential privacy: A survey of results." *International conference on theory and applications of models of computation*. Springer, Berlin, Heidelberg, 2008.

RESPONSE:

We thank the reviewer for this observation and providing the associated references. To assuage the concern, we have now added the following text in the discussion section of our manuscript to provide

specific references for threats that remain for FL and also discuss some possible solutions for the interested reader.

Note that this specific study focused on collaborating sites with an inherent amount of trust, i.e., signed certificates were issued only after pointed discussion between principal investigators of the various collaborating sites. "Open" federations would unavoidably need to deal with a different level of threat and would therefore have different requirements. We have now added the following text in the manuscript towards addressing the reviewer's comment in the Discussion section (page 14, line 581):

Newly added text: *Although data are always retained within the acquiring site during FL (and hence FL is defined as private-by-design), different security and privacy threats remain [1, 2, 3]. These threats include attempted extraction of training data information from intermediate and final models, model theft, and submission of poison model updates with the goal of introducing unwanted model behavior (including incentivizing the model to memorize more information about the training data in support of subsequent extraction, i.e., leakage). A number of technologies can be used to mitigate security and privacy concerns during FL [1, 2, 3]. Homomorphic encryption [4], secure multiparty compute [5], and trusted execution environments (TEEs) [6,7] allow for collaborative computations to be performed with untrusted parties, while maintaining confidentiality of the inputs to the computation. Differentially private training algorithms [8, 9,10] allow for mitigation of information leakage from both the collaborator model updates and the global consensus aggregated models. Finally, assurance that remote computations are executed with integrity can be designed for with the use of hardware-based trust provided by TEEs, as well as with some software-based integrity checking such as those used in [11]. Each of these technologies comes with its own benefits in terms of security and/or privacy, as well as costs and limitations, such as increased computational complexity, associated hardware requirements and/or reduced quality of computational output (such as the reduction of model utility that can be associated with differentially private model training). Further experimentation needs to be done in order to best inform prospective federations as to which technologies to use towards addressing their specific concerns within the context of the collaborator infrastructure and trust levels, depending on the use-case, the extent of the collaborating network, and the level of trust within the involved parties. Our study was based on a collaborative network of trusted sites, where authentication was based on personal communication across collaborating sites and the combination of TLS and TEEs were considered sufficient.*

- [1] Kairouz, Peter, et al. "Advances and open problems in federated learning." Foundations and Trends® in Machine Learning 14.1–2 (2021): 1-210.
- [2] Nasr, Milad, Reza Shokri, and Amir Houmansadr. "Comprehensive privacy analysis of deep learning: Passive and active white-box inference attacks against centralized and federated learning." 2019 IEEE symposium on security and privacy (SP). IEEE, 2019.
- [3] Lam, Maximilian, et al. "Gradient disaggregation: Breaking privacy in federated learning by reconstructing the user participant matrix." International Conference on Machine Learning. PMLR, 2021.
- [4] Gentry, Craig. "Fully homomorphic encryption using ideal lattices." Proceedings of the forty-first annual ACM symposium on Theory of computing. 2009.
- [5] Yao, Andrew C. "Protocols for secure computations." 23rd annual symposium on foundations of computer science (sfcs 1982). IEEE, 1982.

- [6] Sabt, Mohamed, Mohammed Achemlal, and Abdelmadjid Bouabdallah. "Trusted execution environment: what it is, and what it is not." 2015 IEEE Trustcom/BigDataSE/ISPA. Vol. 1. IEEE, 2015.
- [7] Schneider, Moritz, et al. "SoK: Hardware-supported Trusted Execution Environments." arXiv preprint arXiv:2205.12742 (2022).
- [8] Dwork, Cynthia. "Differential privacy: A survey of results." International conference on theory and applications of models of computation. Springer, Berlin, Heidelberg, 2008.
- [9] Wei, Kang, et al. "Federated learning with differential privacy: Algorithms and performance analysis." IEEE Transactions on Information Forensics and Security 15 (2020): 3454-3469., APA,
- [10] Adnan, Mohammed, et al. "Federated learning and differential privacy for medical image analysis." Scientific reports 12.1 (2022): 1-10.
- [11] Tramer, Florian, and Dan Boneh. "Slalom: Fast, verifiable and private execution of neural networks in trusted hardware." arXiv preprint arXiv:1806.03287 (2018).

I firmly believe that these issues of privacy are vitally important to consider in federated learning, and many efforts in the research community have been made in the community to address them (e.g. applications of differential privacy [4], many). I do think it is a detriment to this work that no more details on privacy-preserving efforts are given, and I do think inclusion of some more detail on that front is necessary. I also believe that this further demonstrates that this work does not serve as a blueprint (privacy issues are especially nuanced!), and would like to reiterate my request that the authors tone down that claim. A sufficient blueprint would also need to do more work to address issues in federated learning such as stability to collaborator dropout, bottlenecks in communication between sites, financial considerations, and much more which this work does not.

RESPONSE:

We thank the reviewer for their comment and in retrospect we agree with the given comment. As clarified in our previous response (to the reviewer's second comment) we have now toned down this claim.

Specifically, building upon the text we have added to address the previous comment, we have now removed the word 'blueprint' and instead addressed the ways in which our federation was different (trust among collaborators) and what additional considerations may be needed for other future studies. We have now changed the appropriate text in the Discussion section (page 15, line 623) of our manuscript

Current excerpt (with additions underlined): ... *This study is meant to be used as an example for future FL studies between collaborators with an inherent amount of trust that can result in clinically deployable ML models. ...*

To move away from critical discussion - I do think the scope of the experiments and the work done are strong, and the study stands on its own as a particularly interesting application. The scope and diversity of the study is impressive, and the potential benefits to the research community are clear. I think, though, that this work would benefit more from taking pride in being a wonderfully fascinating case-study, and a marvelous exercise in large-scale federated learning, rather than a generic blueprint which future studies ought to follow - such a claim is simply beyond the scope of this otherwise strong work. In future work, I would encourage the authors to dig more into why certain choices are made during the federated methodology, and to justify why those

decisions were made clear. I would also like to see more detailed discussion on how the administrative tasks were handled - I do firmly believe these insights would significantly benefit the community as a whole by providing a dimension of collaboration which is often omitted from technical papers.

RESPONSE:

We thank the reviewer for the positive comments, and we hope our responses and edits in the manuscript have satisfactorily addressed all the raised concerns. Of note, we also appreciate the view of contributing to our community by offering insights of this study that others have not faced yet, simply because of the study's scale. This was the primary reason that we originally included a description of the administrative tasks we had to overcome to materialize this large-scale real-world study. We look forward to more large-scale future studies sharing insights that we could all benefit as a community and push the frontiers of this field faster towards real clinical/social impact.

To specifically address the suggestion of a more detailed discussion on the administrative tasks, we have now added in the Discussion section (page 13, line 543) the following:

Current excerpt (with additions underlined): Finally, we held interactive sessions to complement the theoretical definition of the reference standards, and further guide collaborating sites. Particular pain points regarding these administrative tasks included managing the large volume of communication (i.e., emails and conference calls) needed to address questions and issues that arose, as well as the downtime incurred in FL training due to issues that had not yet been identified and were adversely affecting the global model. Though we developed many ad-hoc tools for this workflow ourselves (particularly for the data processing and orchestration steps), many issues we encountered were common enough in retrospect (for example common TLS errors) that mature automated solutions will address them. Many of these automations will be use-case dependent, for example the MRI data corruption checks we used from the FeTS library. For these, the associated tools are expected to become available as various domain experts enter into the FL community, while some will be more general purpose. As our inspection of both local and global model validation scores was manual during our deployment, we in retrospect see great value in automated notifications (performed at the collaborator infrastructure to help minimize data information leakage) to alert a collaborator (or the governor) when their local or global model validation is significantly low. Such an alert can indicate the potential need to visually inspect example failure cases in their data for potential issues. With continued efforts towards developing automated administration tools around FL deployments, we expect administration for large FL deployments to become easier.

REVIEWER #3

In this study titled "Federated Learning Enables Big Data for Rare Cancer Boundary Detection," the authors use FML from 71 healthcare institutions across 6 continents to generate an automatic tumor boundary detector for glioblastoma. They report a 33% increased improvement over a publicly trained model to delineate surgical targetable tumor. This is a very comprehensive study with a large amount of data showing the utility of FML.

RESPONSE:

We thank the reviewer for appreciating our study and the complexity of the task.

1. Why was glioblastoma chosen? The rarity is not as rare as other such diseases so not sue if the FML will be as applicable as to say something like an endolymphatic sac tumor, or even low grade glioma for that matter.

RESPONSE:

We thank the reviewer for this comment.

Glioblastoma was selected as the current use-case, taking into consideration, our existing efforts on their delineation through leading the organization of the Brain Tumor Segmentation (BraTS) challenge [1,2,3,4], as well as our previous work [5,6] on identifying the optimal settings for this particularly complex task.

Indeed, there are even more rare diseases than glioblastoma. However, glioblastoma is much rarer comparing to major cancers that it will take some institutions over several years to get to the minimum number we required (100) participating institutions to identify for inclusion in this study. While we could select even more rare primary brain tumor types, the scale of collaborating sites would need to be several hundreds of sites, and at that scale IT infrastructure of the sites and the administrative/coordinating effort would become the bottleneck rendering the completion of this study infeasible. Particularly considering that this was the first study at this scale and complexity, and there was not related literature available to guide us on the challenges we did encounter. There seems to be ongoing efforts to automate parts of the coordination and we would appreciate their materialization towards facilitating further large-scale studies towards expediting research and discovery in healthcare, and addressing clinical requirements. Thus, with the goal of getting to such larger scale eventually to study even more rare diseases, we are hoping to optimize the current platform by building on interoperability with other software tools and automating related processes, thereby enabling easier adoption by more sites with basic IT set-up.

- [1] Menze, Bjoern H., et al. "The multimodal brain tumor image segmentation benchmark (BRATS)." *IEEE transactions on medical imaging* 34.10 (2014): 1993-2024.
- [2] Bakas, Spyridon, et al. "Advancing the cancer genome atlas glioma MRI collections with expert segmentation labels and radiomic features." *Scientific data* 4.1 (2017): 1-13.
- [3] Bakas, Spyridon, et al. "Identifying the best machine learning algorithms for brain tumor segmentation, progression assessment, and overall survival prediction in the BRATS challenge." *arXiv preprint arXiv:1811.02629* (2018).
- [4] Baid, Ujjwal, et al. "The rsna-asnr-miccai brats 2021 benchmark on brain tumor segmentation and radiogenomic classification." *arXiv preprint arXiv:2107.02314* (2021).
- [5] Sheller, Micah J., et al. "Multi-institutional deep learning modeling without sharing patient data: A feasibility study on brain tumor segmentation." *International MICCAI Brainlesion Workshop*. Springer, Cham, 2018.
- [6] Sheller, Micah J., et al. "Federated learning in medicine: facilitating multi-institutional collaborations without sharing patient data." *Scientific reports* 10.1 (2020): 1-12.

2. How was the publicly trained model developed? Why not compare to established ML models?

RESPONSE:

We thank the reviewer for these questions.

To address the first question, the public initial model was developed using the identical DL architecture we used for training the FL consensus model (i.e., a 3D UNet with residual connections) and while utilizing the publicly available dataset of the International Brain Tumor Segmentation (BraTS2020) challenge. We have described this in the "Data" section under "Methods" (page 15, line 633).

In terms of comparing with established ML models for this task, we need to note that the primary focus of these existing models is just on improved output performance (as the main focus in typical related academic research), whereas the focus of our study was also on ensuring clinical deployability of our model. This has immediate consequences in the resources required for the execution/inference of these trained models. To be more specific, the output segmentations of most of the existing established ML models for this task are based on an ensemble of multiple models that increases the computational burden [1,2,3,4]. Clinical environments typically have somewhat constrained computational resources, such as the availability of specialized hardware (e.g., DL acceleration cards) and increased memory, which affect the runtime performance of these models and might not even allow their execution. Therefore, to facilitate the clinical deployability of our model we decided to proceed with a single 3D-ResUNet, rather than an ensemble of multiple of them, ensuring a reduced computational burden, when compared with running multiple. We have discussed this in the “Model Runtime in Low-Resource Settings” sub-section under “Methods” in our manuscript (page 20, line 832).

We hope this addresses the reviewer’s comments, but if not we would very much welcome any further suggestion from the reviewer.

- [1] Menze, Bjoern H., et al. "The multimodal brain tumor image segmentation benchmark (BRATS)." IEEE transactions on medical imaging 34.10 (2014): 1993-2024.
- [2] Bakas, Spyridon, et al. "Advancing the cancer genome atlas glioma MRI collections with expert segmentation labels and radiomic features." Scientific data 4.1 (2017): 1-13.
- [3] Bakas, Spyridon, et al. "Identifying the best machine learning algorithms for brain tumor segmentation, progression assessment, and overall survival prediction in the BRATS challenge." arXiv preprint arXiv:1811.02629 (2018).
- [4] Baid, Ujjwal, et al. "The rsna-asnr-miccai brats 2021 benchmark on brain tumor segmentation and radiogenomic classification." arXiv preprint arXiv:2107.02314 (2021).
- [5] Young, Julio Cristian, and Alethea Suryadibrata. "Applicability of various pre-trained deep convolutional neural networks for pneumonia classification based on X-Ray Images." International Journal of Advanced Trends in Computer Science and Engineering 9.3 (2020).
- [6] Kemker, Ronald, et al. "Measuring catastrophic forgetting in neural networks." Proceedings of the AAAI Conference on Artificial Intelligence. Vol. 32. No. 1. 2018.

3. What about distinguishing and identifying non enhancing glioblastoma?

RESPONSE:

We thank the reviewer for this question.

As we have seen previously, through leading the organization of the International Brain Tumor Segmentation (BraTS) challenge, this deep learning model still identifies the truly non-enhancing glioblastoma (which are generally considered rare, within this rare disease – even more-so now with the diffuse astrocytoma, IDH-mutant, WHO grade 4 tumor type in the WHO 2021 classification criteria [1]) as they would identify the non-enhancing component of enhancing glioblastomas. In other words, for non-enhancing tumors the model will not return any value for the enhancing-tumor label.

- [1] Louis, David N., et al. "The 2021 WHO classification of tumors of the central nervous system: a summary." Neuro-oncology 23.8 (2021): 1231-1251.

4. Where did the FML have the most difficulty?

RESPONSE:

We thank the reviewer for this question.

During our study, we note that the actual coordination and governance of the federation was the most difficult/time-consuming part. We acknowledge that this might be an issue with the specific use-case of our study, and perhaps a classification problem (e.g., binary COVID/non-COVID) could be significantly simpler in terms of data handling. Of note is that this point was also appreciated by Reviewer #2's first and last comments specifically referring to the administrative components of the federation. To collectively address a more detailed discussion on the administrative tasks, we have now added the following in the Discussion section (page 13, line 543)

Current excerpt (with additions underlined): Finally, we held interactive sessions to complement the theoretical definition of the reference standards, and further guide collaborating sites. Particular pain points regarding these administrative tasks included managing the large volume of communication (i.e., emails and conference calls) needed to address questions and issues that arose, as well as the downtime incurred in FL training due to issues that had not yet been identified and were adversely affecting the global model. Though we developed many ad-hoc tools for this workflow ourselves (particularly for the data processing and orchestration steps), many issues we encountered were common enough in retrospect (for example common TLS errors) that mature automated solutions will address them. Many of these automations will be use-case dependent, for example the MRI data corruption checks we used from the FeTS library. For these, the associated tools are expected to become available as various domain experts enter into the FL community, while some will be more general purpose. As our inspection of both local and global model validation scores was manual during our deployment, we in retrospect see great value in automated notifications (performed at the collaborator infrastructure to help minimize data information leakage) to alert a collaborator (or the governor) when their local or global model validation is significantly low. Such an alert can indicate the potential need to visually inspect example failure cases in their data for potential issues. With continued efforts towards developing automated administration tools around FL deployments, we expect administration for large FL deployments to become easier.

=====

REVIEWER #4

=====

There is no doubt that such a global project in medical imaging will enrich our understanding of the benefits of AI for patient care. Thank you to everybody involved.

RESPONSE:

We thank the reviewer for appreciating our study and the positive comment.

Below is a list of my concerns in no particular order:

- DSC is known to deliver higher numbers than IoU (Jaccard index). For a more rigorous evaluation, it would have been much more valuable to report both numbers.

RESPONSE:

We thank the reviewer for this comment.

The primary decision to use DSC instead of IoU was based on the fact that almost all literature related to the evaluation of the use-case of our study is based on DSC, i.e., we note ~200 manuscripts, since 2017 alone, on the very same use-case using DSC for evaluating this task. Below we provide links to all these 2017-2020 papers for reference. Please note that the links to the related 2021 papers (from the BraTS challenge) are not provided yet, as they have not been published yet, but we would like to note that >1,500 international participating teams have submitted results using DSC for evaluating this use-case.

Furthermore, the senior author of this study together with some of the co-authors are part of a consensus effort and manuscripts [1,2], involving experts across 60 institutions worldwide, on metrics appropriate to use for distinct medical imaging tasks. Notably, one of these manuscripts [2] has already received a positive response through a formal full manuscript presubmission in Nature Methods (REDACTED), and is now considered for publication. This effort shows that DSC and Jaccard/IoU are interrelated, with a direct transformation between the two defined by " $\text{IoU} = \text{DSC} / (2 - \text{DSC})$ ". The behavior of these two metrics has been thoroughly evaluated and they are expected to yield the same ranking (of aggregated metric values) in most applications (of course, theoretically, deviations are possible), such that there is no value in combining them, and hence reporting both metrics in a single study is deemed unnecessary. Additionally, there seems to be a domain preference, where the computer vision community prefers the IoU and the medical imaging domain favors DSC.

Taking all the above into consideration, we have chosen to proceed with using the DSC metric instead of the IoU.

2020: (69 papers)

Proceedings Part1: <https://link.springer.com/book/10.1007/978-3-030-72084-1>

Proceedings Part2: <https://link.springer.com/book/10.1007/978-3-030-72087-2>

2019: (57 papers)

Proceedings Part1: <https://link.springer.com/book/10.1007/978-3-030-46640-4>

Proceedings Part2: <https://link.springer.com/book/10.1007/978-3-030-46643-5>

2018 (44 papers)

Proceedings Part1: <https://link.springer.com/book/10.1007/978-3-030-11723-8>

Proceedings Part2: <https://link.springer.com/book/10.1007/978-3-030-11726-9>

2017 (28 papers)

Proceedings: <https://link.springer.com/book/10.1007/978-3-319-75238-9>

[1] Reinke, Annika, et al. "Common limitations of image processing metrics: A picture story." arXiv preprint arXiv:2104.05642 (2021).

[2] Maier-Hein, Lena, et al. "Metrics reloaded: Pitfalls and recommendations for image analysis validation." arXiv preprint arXiv:2206.01653 (2022).

- There is virtually no information on the actual federated learning used. One can only guess that the authors have used the original federated averaging (FedAvg). More details on the federated mechanism would be helpful.

RESPONSE:

We honestly thank the reviewer for pointing this out. We have now added the following text in the Methods section (page 19, line 799) of our manuscript to clarify.

Current excerpt (with additions underlined): ... Each such iteration is called a ``federated round''. Based on our previously conducted performance evaluation for this use-case [2], we chose to perform aggregation of all collaborator updates in the present study, using the federated averaging (FedAvg) approach [3], i.e., average of collaborator's model updates weighted according to collaborator's contributing data. We expect these aggregation strategy choices to be use-case dependent, by providing due consideration to the collaborators' associated compute and network infrastructure.

- [1] Sheller, Micah J., et al. "Multi-institutional deep learning modeling without sharing patient data: A feasibility study on brain tumor segmentation." International MICCAI Brainlesion Workshop. Springer, Cham, 2018.
- [2] Sheller, Micah J., et al. "Federated learning in medicine: facilitating multi-institutional collaborations without sharing patient data." Scientific reports 10.1 (2020): 1-12.
- [3] McMahan, Brendan, et al. "Communication-efficient learning of deep networks from decentralized data." Artificial intelligence and statistics. PMLR, 2017.

- The six sites selected for the out-of-sample generalization test (which is the main motivation for using FL) have been selected due to availability. This is rather a point of concern. Random selection would have been more reliable.

RESPONSE:

We thank the reviewer for this observation, and we completely agree that it would have been ideal to be able for us to randomly select sites for our out-of-sample validation process. However, taking into consideration the global scale of our study and the availability of all collaborating sites (i.e., sites being ready with properly processed and annotated data before starting the federated experiments), we had to devise this (in our opinion) acceptable compromise to keep the scale of our study high, and its successful conclusion in a timely manner, such that it can contribute useful insights to our community. Specifically, if we were waiting for these 6 sites to become ready, then other sites will not have been able to participate and hence either further delay the conclusion of the current study or reduce its scale.

However, beyond identifying the availability-based selection of our out-of-sample validation sites as a limitation, we would like to further point out that one of these 6 sites is the American College of Radiology (ACR), which contributed data from a clinical trial (RTOG0825/ACRIN6686) with 203 participating hospitals. This diverse multi-institutional clinical trial dataset should balance the selection based on availability, and we think that even if the out-of-sample validation was based on random selection for a few sites, ACR should have been intentionally involved as an out-of-sample validation site for the final consensus model, towards assessing its actual generalizability on this large-scale multi-institutional dataset.

- Image samples have not been provided to show the difference between the public initial model and the final consensus model for small, moderate and large differences.

RESPONSE:

We thank the reviewer for the suggestion. We have now provided 3 sample validation cases, from the publicly available BraTS dataset, to qualitatively showcase the segmentation differences across the final global consensus model, the public initial model, and the ground truth annotations. Since these cases are represented by 3D volumetric scans and associated volumetric tumor segmentations, it is difficult to represent and appreciate the extent of segmentation differences in a 2D slice depicted in a manuscript figure. Therefore, to allow the reviewer and the future readers of our manuscript to assess these differences, we have now provided the complete multi-parametric MRI scans and associated segmentation files in a GitHub repository [[https://github.com/FETS-AI/2022 Manuscript Supplement](https://github.com/FETS-AI/2022_Manuscript_Supplement)]. These 3 sample cases denote small, moderate, and large differences between the final consensus model and the public initial model, as we have noted in their respective folder names within the repository. We have further added notes in the repository with recommended open-source visualization tools that can be freely downloaded from the following links:

- CaPTk: <https://www.med.upenn.edu/cbica/captk/>
- FeTS Tool: <https://www.med.upenn.edu/cbica/fets/>
- ITK-SNAP: <http://www.itksnap.org/pmwiki/pmwiki.php>

We have also added this information in the Data Availability section (page 21, line 861):

Newly added text: *Source data are provided with this paper. Specifically, we provide the raw data, the associated python scripts, and specific instructions to reproduce the plots of this study in a GitHub repository, at: [https://github.com/FETS-AI/2022 Manuscript Supplement](https://github.com/FETS-AI/2022_Manuscript_Supplement). The file 'SourceData.tgz', in the top directory, holds an archive of csv files representing that are the source data. The python scripts are provided in the 'scripts' folder which utilize these source data and save .png images to disc and/or print latex code (for tables) to stdout. Furthermore, we have provided 3 sample validation cases, from the publicly available BraTS dataset, to qualitatively showcase the segmentation differences (small, moderate and large) across the final global consensus model, the public initial model, and the ground truth annotations in the same GitHub repository.*

- The data quality issue regarding erroneous reference annotations, or inherent variability, has not been properly addressed, among others, for the validation through six sites. Were the 590 cases delineated by one expert only?

RESPONSE:

The reviewer has posed a very interesting question.

We have communicated to all collaborators the actual processing, harmonization, and annotation protocol that their data need to undergo (please see <https://fets-ai.github.io/Front-End/runningApplication#manual-corrections> for the detailed instructions). We have also provided a software solution using the accompanying FeTS Tool that they could leverage to make this task easier. However, since each individual collaborating site performed their own annotations, and we did not have access to their data, we cannot conclusively speak to the precise mechanism followed at individual locations. We would like to note that the use-case was inherently complex, i.e., it required substantial amount of data processing and manual corrections to automatically generated labels before any training could start. This added a significant burden to the coordination process, in terms

of effectively communicating the requirements, and in a lot of cases actually debugging the process on video conference calls, which was compounded by the fact that collaborators were present across all time-zones. We think that more community-led efforts [1] are needed to enable more stable, automated, and expedited orchestration of a federation, and then allow researchers to focus on the development of more general-purpose and specialized data sanity checks for different problems.

[1] Karargyris, Alexandros, et al. "MedPerf: Open Benchmarking Platform for Medical Artificial Intelligence using Federated Evaluation." arXiv preprint arXiv:2110.01406 (2021).

- A more in-depth and conclusive analysis regarding the comparison of the ensemble approach and the FL network is missing. Would a network of few major hospitals have the same effect as a complicated global network?

RESPONSE:

We thank the reviewer for their insightful question. We feel that the answer is dependent on the quality and diversity of the data held at the ‘few major hospitals’, and therefore can be difficult to know in advance. We see evidence of this in the formal analysis of the preliminary consensus model (which was a network of 35 sites) and the final consensus model (which was a network of 71 sites) that we performed and presented in the section “results” subsection “Data size alone may not predict success”:

(section Results, page 9, line 380) We further expanded this analysis to assess this observation in a non-federated configuration, where we selected the largest collaborating sites (comprehensive cancer centers contributing > 200 cases, and familiar with computational analyses), and coordinated independent model training for each, starting from the public initial model and using only their local training data. The findings of this evaluation indicate that the final consensus model performance is always superior or insignificantly different ($p_{Average} = 0.1$, $p_{ET} = 0.5$, $p_{TC} = 0.2$, $p_{WT} = 0.06$, Wilcoxon signed-rank test) to the ensemble of the local models of these four largest contributing collaborators, for all tumor sub-compartments (Fig. 2). This finding highlights that even large institutions can benefit from collaboration.

We have performed the entire performance analysis based on the current literature [1,2]. However, we concede that we have no control in the way the ground truth was created at collaborating sites (data quality) nor do we have control over the unique characteristics of the scans at each site (data diversity). Even though the preliminary consensus model performed similar to the final consensus model, we note that the final consensus was demonstrably better. Though this increased performance is not statistically significant ($p > 0.067$, Wilcoxon signed rank test), it does indicate that larger institutions can still benefit from collaborating with smaller ones. Such a benefit could be due to the distribution of the patient population (e.g., demographics) and imaging characteristics (e.g., acquisition protocols) at the smaller sites, despite the fact that they hold fewer cases. We have made the following change in the Discussion section (page 12, line 479) of the manuscript to further highlight that even institutions with large patient cohorts can still benefit from collaboration.

Current excerpt (with additions underlined): *To further assess these considerations, we coordinated independent model training for the four largest collaborating sites (i.e., > 200 cases) by starting from the same public initial model and using only their local training data. The ensemble of these four largest site local models did not show significant performance*

differences to the final consensus model for any tumor sub-compartment, yet the final consensus model showed superior performance indicating that even institutions with large datasets can benefit from collaboration. The underlying assumption for these results is that since each of these collaborators initiated their training from the public initial model (which included diverse data from 16 sites), their independent models and their ensemble have inherited some of the initial model's data diversity, which could justify the observed insignificant differences (Fig. 2 and Fig. 6).

[1] Reinke, Annika, et al. "Common limitations of image processing metrics: A picture story." arXiv preprint arXiv:2104.05642 (2021).

[2] Maier-Hein, Lena, et al. "Metrics reloaded: Pitfalls and recommendations for image analysis validation." arXiv preprint arXiv:2206.01653 (2022).

- Why were you not able to evaluate the exclusion of problematic sites like 48?

RESPONSE:

We thank the reviewer for this perceptive observation.

During training we were only able to identify major issues that had a direct impact on the convergence curves of the global consensus model, i.e., "global validation score". It seems that since the data contributions of the site 48 were too small (n=46) when compared with the total data included from the whole federation (n=6,314), their problematic inclusion had only a minimal impact in the convergence curves and hence was not caught during training. However, we found out about this problematic site after the study was done, at which point it was too late to exclude them from the analysis, and in order to do so it would require a complete re-initialization of the whole FL study.

This issue reflects the importance of governance solutions that could identify such issues during training, or even prior to initiating such studies by assessing statistical/distribution characteristics of the contributing data from each collaborating site and their relation to the overall data involved from all collaborators. We have added a specific paragraph in our manuscript ("Results" section: "FL is robust to data quality issues", page 9, lines 389) to address this point which reads as follows:

Data quality issues relating to erroneous reference annotations (with potential negative downstream effect on output predictions) were identified by monitoring the global consensus model performance during training. However, only data quality issues that largely affected the global validation score could be identified and corrected during training. Those with more subtle effects in the global validation score were only identified after the completion of the model training, by looking for relatively low local validation scores of the consensus model across collaborating sites.

To further address this reviewer's comment, collectively with the other reviewers' points about adding a more detailed discussion on the administrative tasks of the study, we have now added the following text (Section Discussion, page 13, line 543)

Current excerpt (with additions underlined): Finally, we held interactive sessions to complement the theoretical definition of the reference standards, and further guide collaborating sites. Particular pain points regarding these administrative tasks included managing the large volume of communication (i.e., emails and conference calls) needed to address questions and issues that arose, as well as the downtime incurred in FL training due to issues that had not yet been

identified and were adversely affecting the global model. Though we developed many ad-hoc tools for this workflow ourselves (particularly for the data processing and orchestration steps), many issues we encountered were common enough in retrospect (for example common TLS errors) that mature automated solutions will address them. Many of these automations will be use-case dependent, for example the MRI data corruption checks we used from the FeTS library. For these, the associated tools are expected to become available as various domain experts enter into the FL community, while some will be more general purpose. As our inspection of both local and global model validation scores was manual during our deployment, we in retrospect see great value in automated notifications (performed at the collaborator infrastructure to help minimize data information leakage) to alert a collaborator (or the governor) when their local or global model validation is significantly low. Such an alert can indicate the potential need to visually inspect example failure cases in their data for potential issues. With continued efforts towards developing automated administration tools around FL deployments, we expect administration for large FL deployments to become easier.

- Recent literature has not been mentioned. For instance:

- 1) ProxyFL: Decentralized Federated Learning through Proxy Model Sharing
- 2) Federated Learning with Differential Privacy: Algorithms and Performance Analysis
- 3) Federated learning and differential privacy for medical image analysis
- 4) Federated Learning for Computational Pathology on Gigapixel Whole Slide Images

RESPONSE:

We thank the reviewer for their cognizance in this field. We have now added the new citations in the last paragraph of the “discussion” section (page 15, line 623) including appropriate text:

Current Excerpt (with additions underlined):

We have demonstrated the utility of an FL workflow to develop an accurate and generalizable ML model for detecting glioblastoma sub-compartment boundaries, a finding which is of particular relevance for neurosurgical and radiotherapy planning in patients with this disease. This study is meant to be used as an example for future FL studies between collaborators with an inherent amount of trust that can result in clinically deployable ML models. Further research is required to assess privacy concerns in a detailed manner [1,2] and to apply FL to different tasks and data types as in [3,4,5,6]. Building on this study, a continuous FL consortium would enable downstream quantitative analyses with implications for both routine practice and clinical trials, and most importantly, increase access to high-quality precision care worldwide. Furthermore, the lessons learned from this study with such a global footprint are invaluable and can be applied to a broad array of clinical scenarios with the potential for great impact to rare diseases and underrepresented populations.

[1] Wei, Kang, et al. "Federated learning with differential privacy: Algorithms and performance analysis." *IEEE Transactions on Information Forensics and Security* 15 (2020): 3454-3469.

[2] Adnan, Mohammed, et al. "Federated learning and differential privacy for medical image analysis." *Scientific reports* 12.1 (2022): 1-10.

[3] Kalra, Shivam, et al. "ProxyFL: Decentralized Federated Learning through Proxy Model Sharing." *arXiv preprint arXiv:2111.11343* (2021).

[4] Lu, Ming Y., et al. "Federated learning for computational pathology on gigapixel whole slide images." *Medical image analysis* 76 (2022): 102298.

- [5] Baid, Ujjwal, et al. "Federated Learning for the Classification of Tumor Infiltrating Lymphocytes." *arXiv preprint arXiv:2203.16622* (2022).
- [6] Linardos, Akis, et al. "Federated learning for multi-center imaging diagnostics: a simulation study in cardiovascular disease." *Scientific Reports* 12.1 (2022): 1-12.

Reviewers' Comments:

Reviewer #3:

Remarks to the Author:

All issues addressed

Reviewer #4:

Remarks to the Author:

Thank you for making changes. However, it is perplexing to me why you are resiting to add IoU (Jaccard) beside Dice values. You would not even need to rerun any major experiments to add IOU values.